# WHICH REWARDS MATTER?
# REWARD SELECTION FOR REINFORCEMENT LEARNING UNDER LIMITED FEEDBACK

## ABSTRACT

The ability of reinforcement learning algorithms to learn effective policies is determined by the rewards available during training. However, for practical problems, obtaining large quantities of reward labels is often infeasible due to computational or financial constraints, particularly when relying on human feedback. When reinforcement learning must proceed with limited feedback—only a fraction of samples get rewards labeled—a fundamental question arises: *which* samples should be labeled to maximize policy performance? We formalize this problem of *reward selection* for reinforcement learning from limited feedback (RLLF), introducing a new problem formulation that facilitates the study of strategies for selecting impactful rewards. Two types of selection strategies are investigated: (i) heuristics that rely on reward-free information such as state visitation and partial value functions, and (ii) strategies pre-trained using auxiliary evaluative feedback. We find that critical subsets of rewards are those that (1) guide the agent along optimal trajectories, and (2) support recovery toward near-optimal behavior after deviations. Effective selection methods yield near-optimal policies with significantly fewer reward labels than full supervision, establishing reward selection as a powerful paradigm for scaling reinforcement learning in feedback-limited settings.

## 1 INTRODUCTION

Various real-world scenarios of sequential decision-making share a striking asymmetry: while data is abundant (or cheaply generated), obtaining evaluative feedback is prohibitively costly and therefore limited by practical constraints. Consider the following examples: in reinforcement learning from human feedback (RLHF) for training large language models (LLMs), billions of tokens can be generated easily, but acquiring reliable human feedback carries significant operational overhead (Christiano et al., 2017; Ouyang et al., 2022; Bai et al., 2022; ABAKA AI, 2025). In the field of AI-driven drug discovery, modern generative models can enumerate billions of syntactically valid molecular graphs in silico, sweeping through an estimated chemical space of $\approx 10^{60}$ drug-like molecules (Reymond, 2015; Gómez-Bombarelli et al., 2018; Jin et al., 2019). Yet confirming that any one of those structures is synthesizable, binds to the intended target, and is non-toxic requires weeks of wet-lab assays and thousands of dollars per compound (DiMasi et al., 2016; Anon, 2023). In these and many similar problems (Appendix A), where evaluative feedback is limited, it becomes critical to identify which subset of the abundant data should be selected for feedback in order to achieve maximal performance gain with minimal feedback.

Reinforcement learning (RL) is the widely adopted approach for solving sequential decision-making problems (Popova et al., 2018; Ouyang et al., 2022; Feng et al., 2023). In the RL framing of the above scenarios, feedback corresponds to rewards, but obtaining rewards for all data points is infeasible. In this work, we study the important question of *reward selection*—which subset of the data should be labeled with rewards to maximize the performance of the learned policy? Acquiring rewards for different subsets leads to policies of varying quality, and the goal is to select the parts of the dataset to be reward-labeled such that the resulting policy achieves the highest performance, as illustrated in Figure 1. The question of which data points to acquire rewards for is equivalent to selecting the states at which to observe rewards. Consequently, the problem is formulated as the selection of a subset of states at which to obtain rewards. We formulate the reward selection problem wherein the

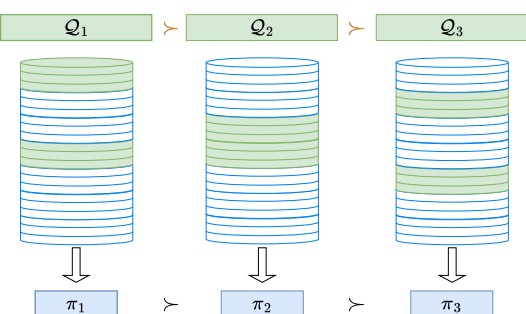

Figure 1: Each row represents a data sample; shaded green rows indicate samples that have been labeled with rewards. The strategy $\mathcal{Q}_i$ determines which states to select for reward labeling. In the limited feedback setup, only a subset of states can be labeled. Different choices of reward-labeled subsets yield learnt policies of varying performances. The objective is to identify the subset that leads to the highest-performing policy.

only degree of freedom permitted is the selection of states (as input), and the outcome observed is the resultant policy (as output), as illustrated in Figure 2 and detailed in Section 2.2.

The reward selection strategies studied, by design, are agnostic to the specifics of the reinforcement learning under limited feedback (RLLF) methodology—particularly the reward generation protocols—allowing the formulation and analysis to generalize to future methods of reward generation. Furthermore, we consider RLLF on offline datasets to disentangle the conflating effects of online state reachability and exploration. That is, any selected states can be labeled with rewards for training, rather than only those than can be reached by an exploration policy. This contrasts with prior setups within active RL (Krueger et al., 2020) and partially observable rewards (Parisi et al., 2024b), which share similar motivations. To learn from partially reward-labeled data, we adapt an existing algorithm for incorporating unlabeled data with labeled data for (offline) RL (Yu et al., 2022). Alternatively, we also study a variant of Q-learning (in Appendix D.9) that defaults to imitating the data-collecting policy on unlabeled data.

We begin by developing evaluating a range of heuristic selection strategies, including one that adaptively balances between two heuristics (Section 3.1). Their effectiveness depends strongly on domain traits, which we characterize in Section 4.1. For cases where feedback about the performance of intermediate policies is obtainable, we propose a training-phase formulation in which selection strategies themselves can be optimized (Section 3.2); using methods like evolutionary search, we study how such strategies improve with additional training cost and compare them to heuristic approaches (Section 4.2). Finally, we analyze the best (optimal) reward selections to identify structural patterns that explain which rewards matter most under limited feedback (Section 4.3). Effective reward selection yields near-optimal policies with far fewer reward labels than full supervision, highlighting both the potential and challenges of feedback-efficient reinforcement learning.

In this work, **our contributions** are:

1. Formulate the problem of *reward selection* for reinforcement learning under limited feedback, establishing a general, domain-agnostic framework with practical relevance across diverse applications such as RLHF for LLMs and AI-driven drug discovery (see Appendix A).

2. Conduct a systematic investigation of the problem landscape by developing and evaluating a range of heuristic-based strategies, characterizing how different design principles influence downstream policy performance.

3. Introduce a training-phase optimization setting where selection strategies themselves can be trained from feedback, illustrating how data-driven approaches compare to heuristic ones at the cost of additional training.

4. Provide an analysis of optimal reward selections, revealing structural factors that answer the central question: *which rewards matter?*—laying the groundwork for future algorithmic development.

## 2 PROBLEM FORMULATION AND PRELIMINARIES

**Preliminaries:** An MDP is a tuple $M := (\mathcal{S}, \mathcal{A}, p, r, \gamma, \eta)$ where $\mathcal{S}$ is a finite set of states, $S_t$ is the state at time $t \in \{0, 1, \dots\}$, $\mathcal{A}$ is a finite set of actions, $A_t$ is the action at time $t$, $p : \mathcal{S} \times \mathcal{A} \times \mathcal{S} \to [0, 1]$ is the *transition function* that characterizes state transition dynamics according to $p(s, a, s') := \Pr(S_{t+1}=s'|S_t=s, A_t=a)$, $r : \mathcal{S} \times \mathcal{A} \to \mathbb{R}$ is the *reward function* that characterizes rewards according to $r(s, a) := \mathbb{E}[R_t|S_t=s, A_t=a]$, $\gamma \in [0, 1]$ is the reward discount parameter, and $\eta : \mathcal{S} \to [0, 1]$ characterizes the initial state distribution according to $\eta(s) := \Pr(S_0=s)$. A policy

$\pi : \mathcal{S} \times \mathcal{A} \to [0,1]$ characterizes how actions can be selected given the current state according to $\pi(s, a) := \Pr(A_t{=}a|S_t{=}s)$. We consider finite horizon MDPs (Sutton & Barto, 2018) where episodes terminate by some (unspecified) time $T \in \mathbb{N}$.

## 2.1 REINFORCEMENT LEARNING FROM LIMITED FEEDBACK

We study the problem of reinforcement learning from limited feedback (RLLF) in the offline setting. An offline dataset $\mathcal{D}_n = \{(S_t, A_t, S_{t+1})^{(i)}\}_{i=1}^n$ of $n$ samples is obtained by the interaction of a *data-collecting policy* $\pi_D$ with $M$.[1] The dataset contains no reward, i.e., evaluative feedback. To emulate the limited feedback setting, the restriction imposed by the problem setup is that environment rewards are permitted to be obtained at only a subset $B$ of the states. Let $\mathcal{S}_{[B]}$ denote the states that are reward-labeled. For samples in $\mathcal{D}$ where $S_t \in \mathcal{S}_{[B]}$, reward labels are assigned; the remaining samples in $\mathcal{D}$ are unlabeled. In practice, since the *labeling budget* is smaller than the total number of states $|\mathcal{S}|$, only a subset of the dataset can be reward-labeled. The process of reward-labeling part of the dataset and learning a policy from the resulting partially labeled data is referred to as reinforcement learning from limited feedback, and is denoted by $\text{RLLF}(\mathcal{D}, \mathcal{S}_{[B]})$ (see the box in Figure 2). Different choices of $\mathcal{S}_{[B]}$ result in different policies learned from the partially labeled dataset, with varying performance (see Figure 6 in Appendix D.2).

Rather than passively learning a policy from a given partially labeled dataset, we study the problem of actively selecting the states to label with rewards in order to obtain the best-performing policy. Formally, the **reward selection** problem is to *identify a subset of states $\mathcal{S}_{[B]}$, subject to a labeling budget $B$, to be labeled with rewards such that the policy learned from the resulting partially labeled dataset achieves maximum performance.*

**Policy Learning from Partially Reward-Labeled Data:** Given a dataset where only a subset of samples are reward-labeled, we use the UDS algorithm (Yu et al., 2022) to learn a policy from the partially reward-labeled dataset. This algorithm follows a simple procedure: unknown rewards are replaced with zero (or $R_{\min}$), and a policy is learned using these imputed rewards. We adopt Q-learning as the policy update rule, as is standard in offline RL settings (Levine et al., 2020; Kostrikov et al., 2021). Other methods for handling partially labeled data could also be employed, but the focus of this work is on identifying a reward selection strategy that is effective for this instantiation of RLLF. An alternative policy learning rule, which sets the Q-values of states with unknown rewards to zero, is also studied in Appendix D.9.

## 2.2 REWARD SELECTION

The strategy for selecting the $B$ states from $\mathcal{D}$ to label with rewards is denoted by $\mathcal{Q}^{(B)} : \mathcal{D} \to \mathcal{S}^B$. Formally, given a budget $B$, the set of states at which rewards are observed is defined as $\mathcal{S}_{[B]} = \mathcal{Q}^{(B)}(\mathcal{D})$. The resulting policy is denoted by $\pi_{[B]} = \text{RLLF}\left(\mathcal{D}, \mathcal{Q}^{(B)}(\mathcal{D})\right)$.[2] The effectiveness of a strategy $\mathcal{Q}^{(B)}$ is quantified by the expected return of the policy produced by RLLF when trained using the rewards selected by $\mathcal{Q}^{(B)}$. The objective, denoted by $P(\cdot)$, is to maximize the *average* expected return of the resulting policy, $J(\pi) := \mathbb{E}_\pi \left[\sum_{t=0}^T \gamma^t R_t\right]$, averaged over possible datasets $\mathcal{D}$. That is,

$$\max_{\mathcal{Q}^{(B)}} P(\mathcal{Q}^{(B)}) := \max_{\mathcal{Q}^{(B)}} \mathbb{E}_\mathcal{D} \left[ J\left(\pi_{[B]}\right) \right] = \max_{\mathcal{Q}^{(B)}} \mathbb{E}_\mathcal{D} \left[ J\left( \text{RLLF}\left(\mathcal{D}, \mathcal{Q}^{(B)}(\mathcal{D})\right)\right)\right]. \quad (1)$$

When $\mathcal{Q}^{(B)}$ is stochastic, the definition of $P(\cdot)$ includes an additional nested expectation over $\mathcal{Q}^{(B)}$.

**Optimality:** Given a budget $B$, the *optimal* reward selection strategy $\mathcal{Q}^{(B)}$ maximizes the performance of the resultant policy $\pi_{[\mathcal{S}_{[B]}]}$. There are $\binom{|\mathcal{S}|}{B}$ candidate state sets that may be chosen by $\mathcal{Q}^{(B)}$, all resulting in varying policies with varying performances (Appendix D.2). The optimal strategy

---

[1]The data-collecting policy $\pi_D$ can be a single policy, or a mixture of policies of which the weighted average is denoted by $\pi_D$. For clarity, we drop the subscript $n$ unless explicitly needed, and denote the dataset by $\mathcal{D}$.

[2]To make the dependence on $\mathcal{S}_{[B]}$ explicit, $\pi_{[B]}$ is equivalently denoted as $\pi_{[\mathcal{S}_{[B]}]}$ when relevant.

entails selecting a state set, denoted by $\mathcal{S}^*_{[B]}$, that results in a policy with the highest performance, i.e.,

$$\mathcal{S}^*_{[B]} = \arg \max_{\mathcal{S}_{[B]} \subseteq \mathcal{S}, |\mathcal{S}_{[B]}| = B} P(\pi_{[\mathcal{S}_{[B]}]}) = \arg \max_{\mathcal{Q}^{(B)}(\mathcal{D}) \subseteq \mathcal{S}} P(\text{RLLF}(\mathcal{D}, \mathcal{Q}^{(B)}(\mathcal{D}))). \tag{2}$$

It must be noted that $\mathcal{S}^*_{[B]}$ may not be a unique set, rather, it belongs to a set of *equally optimal state sets*. For ease of exposition, we pick one such state set. The efficacy of any other strategy, that selects a different state set $\mathcal{S}_{[B]}$, can be quantified by the *optimality gap*, i.e., the gap from the performance of the optimal policy under the labeling budget $\pi^*_{[B]} = \pi^*_{[\mathcal{S}^*_{[B]}]}$, given by:

$$\text{OptimalityGap}(\mathcal{S}_{[B]}) = P(\pi^*_{[B]}) - P(\pi_{[\mathcal{S}_{[B]}]}). \tag{3}$$

**Setup:** Without insight into how selecting specific states affects final policy performance, it is challenging to design effective reward selection strategies. To enable more informed design, we introduce an optional *training phase* in which the reward selection learner $\mathcal{Q}^{(B)}$ leverages feedback from an evaluator $\Xi$. The evaluator provides the expected return of any policy under the true reward function of $M$, but only at the aggregate level—individual rewards are neither stored nor reused. In practice, this could correspond to deploying a policy trained on limited feedback and using its performance as a signal to refine the reward selection strategy. Once trained, a strategy is evaluated in a *test phase*, where access to $\Xi$ is no longer available. This setup is illustrated in Figure 2.

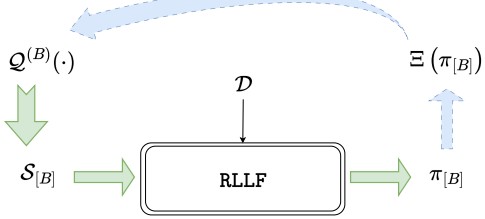

Figure 2: Problem setup for reward selection: The green arrows indicate the test phase, during which the reward selection strategy is evaluated. The blue arrows represent access to, and feedback from, an evaluator available within the training phase loop.

During the **training phase**, $\Xi$ assesses policies induced by different state subsets, guiding updates to the selection strategy. The RLLF procedure is treated as a black box: individual state-reward values and policy update mechanisms remain inaccessible. RLLF takes a set of states and an unlabeled dataset as input and outputs a policy, which may optionally be evaluated by $\Xi$ for training.

During the **test phase**, reward selection strategies are compared along two dimensions: (1) their performance, as defined in Equation 1, and (2) their training cost, measured by the number of calls to $\Xi$. An ideal strategy maximizes test performance while minimizing evaluator usage. Training data $\mathcal{D}_{\text{train}}$ are generated by a data-collecting policy $\pi_{\text{train}}$, while test datasets $\mathcal{D}_{\text{test}}$ come from policies $\Pi_{\text{test}} = \{\pi_1, \ldots, \pi_m\}$. The test performance of $\mathcal{Q}^{(B)}$ is averaged over $\mathcal{D}_{\text{test}}$, as in Equation 1.

## 3 METHODOLOGY: SELECTION STRATEGIES

We study two types of selection strategies. The first category consists of strategies guided by intuitive heuristics that are rule-based and do not rely on the training phase. Thus, they can be expected to perform well enough, though not optimally. Their primary purpose is to assess the utility of intuitive heuristics when applied to the problem of reward selection without access to any prior information. The second category includes strategies that incorporate a training phase prior to evaluation. Within this category, we study a spectrum of approaches: from strategies that identify the optimal reward-labeled state set $\mathcal{S}^*_{[B]}$, albeit at high training cost, to approximate strategies that reduce training overhead at the expense of marginal loss in performance. Additionally, the strategies we study can be classified based on how they construct the reward-labeled state set: *batch strategies*, which select all $B$ states at once, and *iterative strategies*, which select one state at a time over $B$ iterations. Iterative strategies are indexed by $b \in 1, \ldots, B$, with selected states and related quantities indexed by $b$, for instance the set of selected states $\mathcal{S}_{[b]}$. A detailed categorization is provided in Appendix C.

### 3.1 HEURISTIC-BASED SELECTION: TRAINING-FREE STRATEGIES

Given an offline dataset $\mathcal{D}$, without any feedback to inform how labeling different states with rewards impacts the performance of the policy, we must rely on heuristics to guide our selection of states to

label with rewards. The state-visitation distribution of the data collecting policy $\pi_D$, captured within the offline test dataset, serves as a useful signal to guide the selection of states for reward-labeling. Additionally, constructing the state set (of size $B$) iteratively, i.e., adding one state at a time, allows for intermediate updates (at iteration $b$) to the policy and the corresponding Q-values to inform subsequent selections. The heuristics investigated are:

(1) `visitation` sampling: This strategy encodes the intuitive notion that maximizing the fraction of the dataset that is reward-labeled is a good proxy for maximizing the expected return of the resultant policy. To do so, it samples the most commonly occurring states in the dataset without replacement from the state-visitation distribution $d^{\pi_D}$, i.e., $\mathcal{S}_{[B]} \sim \text{Sample}_{\text{w/o rep}}(\mathcal{S}, d^{\pi_D}, B)$.

    (a) If $\mathcal{S}_{[B]}$ is constructed in an *iterative* manner, i.e., adding one state at a time, as opposed to a *batch* manner as above, an additional *on-policy* variant of this strategy is studied, referred to as `visitation-on-policy`, where the state set $\mathcal{S}_{[B]}$ is constructed by sampling states from the state-visitation distribution of the updated policy $\pi_{[b-1]}$ at each iteration $b$.

(2) `uniform` sampling: This simple strategy samples $B$ states without replacement from a uniform distribution over all unlabeled states, i.e., $\mathcal{S}_{[B]} \sim \text{UniformSample}_{\text{w/o rep}}(\mathcal{S}, B)$. Along with serving as a baseline for comparison with other strategies, this simple strategy turns out to be surprisingly effective in certain cases where states that are not frequently visited under $\pi_D$ can have high utility when labeled with rewards.

(3) `guided` sampling : This is an iterative strategy that balances exploration and exploitation—by exploring via sampling from the state-visitation distribution, and exploiting by sampling from the neighborhood of the current highest valued state. Specifically, at each iteration $b$, the strategy samples from the distribution $q_b$ defined as:

$$q_b(\cdot|Q^{\pi_{[b-1]}}, b) \propto \alpha_b \underbrace{d^{\pi_D}(\cdot)}_{\text{explore}} + (1 - \alpha_b) \underbrace{d^{\pi_D}_{\text{prev}}(\cdot \mid \arg\max_{s \in \mathcal{S}} \max_{a \in \mathcal{A}} Q^{\pi_{[b-1]}}(s, a))}_{\text{exploit: focus on states near the most promising Q-values}} \tag{4}$$

where $\widehat{d}^{\pi_D}_{\text{prev}}(\cdot \mid s')$ is the sample estimate of the distribution of states that lead to state $s'$ as the next state under $\pi_D$. The term $\arg\max_{s \in \mathcal{S}_{[b-1]}} \max_{a \in \mathcal{A}} Q^{\pi_{[b-1]}}(s, a)$ identifies the state with the maximum (state-)value based on the rewards obtained thus far. The tradeoff weight $\alpha_b$ initially places more weight on the exploratory term and then decays as $b$ increases, with decreasing $\alpha_b$ as Q-values become more reliable.

    (a) The on-policy variant of this strategy, `guided-on-policy`, is also studied.

We estimate the state visitation distribution(s) $d^{\pi_D}(\cdot)$ from the dataset $\mathcal{D}$, denoted by $\widehat{d}^{\pi_D}(\cdot)$, as $\widehat{d}^{\pi_D}(s) := \frac{N(s)}{\sum_{s' \in \mathcal{S}} N(s')}$, where $N(s)$ denotes the number of occurrences of state $s$ in $\mathcal{D}$. For the on-policy variants, we construct a maximum likelihood estimate of the transition model directly from the offline dataset by empirically counting the number of transitions, i.e., $\widehat{p}(s, a, s') = N(s, a, s')/N(s, a)$. This estimated model is used to compute the state visitation distribution for the policy under consideration. These strategies are empirically evaluated in Section 4 and compared to the training-based strategies described in the next section.

## 3.2 STRATEGIES LEVERAGING THE TRAINING PHASE

For the set of strategies that leverage the training phase, the feedback from the evaluator provides a key insight: the impact of the selected states on the performance of the resultant policy, and, consequently, the performance of the strategy (Equation 1). The selected set of states can subsequently be updated to improve the performance of the resultant policy. The cost of this training phase, prior to the strategy's evaluation, is quantified by the number of calls to the evaluator $\Xi$. This cost captures the operational costs involved in evaluating a policy in settings where obtaining evaluative feedback is resource-intensive.

(1) The most straightforward strategy is to exhaustively search over all possible subsets of $B$ states during the training phase, and select the one that results in the highest performing policy. This approach, referred to as `brute-force`, is guaranteed to find the optimal state set $\mathcal{S}^*_{[B]}$, given sufficient coverage of the training data. However, since the number of all possible subsets of size

$B$ that must be evaluated is *combinatorially* large—$\binom{|\mathcal{S}|}{B} \approx O(|\mathcal{S}|^{\min\{|\mathcal{S}|-B,B\}}) \approx O(|\mathcal{S}|^B)$—the resulting **training cost** is impractical for any reasonably sized state space $\mathcal{S}$.

(2) To mitigate the training cost, we investigate an iterative strategy that constructs the state set $\mathcal{S}_{[B]}$ one state at a time. Specifically, define the *utility* of adding $s$ to $\mathcal{S}_{[b]}$ as

$$\Delta(s|\mathcal{S}_{[b]}) := P(\pi_{[\mathcal{S}_{[b]} \cup \{s\}]}) - P(\pi_{[\mathcal{S}_{[b]}]}). \tag{5}$$

The `sequential-greedy` strategy selects the state $s$ that maximizes $\Delta(s|\mathcal{S}_{[b]})$, i.e., the marginal utility of adding state $s$ to the current set of states $\mathcal{S}_{[b]}$ at each iteration $b$. As a result, this strategy has a **training cost** of $O(B|\mathcal{S}|)$, significantly lower than the brute force strategy. Furthermore, we empirically observe that the sequential-greedy strategy is approximately optimal in many cases.

(3) Lastly, instead of relying on a rule-based approach, we optimize the selection strategy $\mathcal{Q}^{(B)}$ using an evolutionary strategy (ES) (Rechenberg, 1989; Salimans et al., 2017). We parameterize the selection strategy $\mathcal{Q}^{(B)}$ with parameters $\theta$, i.e., $\mathcal{Q}_{\theta}^{(B)}$. We define the fitness of each state set $\mathcal{S}_{[b]}$ as the performance of the resulting policy $J(\pi_{[\mathcal{S}_b]})$, and run a few iterations of ES to optimize $\theta$. The population $k$ in each iteration of ES, and the number of iterations $m$, determine the overall **training cost** $O(km)$ of this strategy, referred to as `ES`.

A categorization of all selection strategies is provided in Appendix C.1.

## 4 EMPIRICAL ANALYSIS

This section evaluates the reward selection strategies across diverse domains. We empirically investigate the following questions: **(Q1**: Section 4.1**)** Which heuristics are effective proxies for reward selection and what factors shape their effectiveness? **(Q2**: Section 4.2**)** What performance benefits does a training phase provide given its additional cost? **(Q3**: Section 4.3**)** What characteristics define high-impact rewards whose selection should be prioritized under limited budgets?

**Domains:** We evaluate performance across six prototypical domains and four large-scale MinAtar domains (Young & Tian, 2019) (Breakout, Freeway, Seaquest, Asterix). Of the prototypical domains, some (Graph, Tree, TwoRooms, TwoRooms-Trap) are purpose-built; others (FrozenLake, CliffWalk) are standard Gymnasium benchmarks (Brockman et al., 2016; Foundation, 2023). Additional domain details, transition dynamics, reward structures, expert policies, data collection, and further experiments on TwoRooms-Trap and FrozenLake appear in Appendix D.1.

**Evaluation:** The primary evaluation metric is the **average episode return**, reported across all experiments. For heuristic-based selection, we additionally report the **optimality gap**, as defined in Equation 3. All reward acquisition budgets are expressed as percentage feedback relative to the

Table 1: Comparison of `guided`, `visitation`, and `uniform` heuristic selection strategies on prototypical domains. For each domain, the table presents the optimality gap and the corresponding mean policy return ± standard error (in parentheses) across five feedback levels. Across all strategies and domains, the optimality gap decreases with increasing budget.

| Domains | Percentage Feedback | guided | guided-on-policy | visitation | visitation-on-policy | uniform |
|---|---|---|---|---|---|---|
| Graph | 0.1 | 3.3 [3.7 ± 0.1] | 3.8 [3.2 ± 0.1] | **3.2 [3.8 ± 0.2]** | 3.7 [3.3 ± 0.1] | 4.1 [2.9 ± 0.1] |
| | 0.3 | 2.2 [5.8 ± 0.1] | 2.2 [5.8 ± 0.1] | **2.1 [5.9 ± 0.1]** | 2.3 [5.7 ± 0.1] | 3.4 [4.6 ± 0.2] |
| | 0.5 | 0.9 [7.1 ± 0.1] | **0.3 [7.7 ± 0.1]** | 0.8 [7.2 ± 0.1] | 0.4 [7.6 ± 0.1] | 2.0 [6.0 ± 0.1] |
| | 0.7 | 0.2 [7.8 ± 0.0] | **0.0 [8.0 ± 0.0]** | 0.4 [7.6 ± 0.1] | 0.0 [8.0 ± 0.0] | 1.1 [6.9 ± 0.1] |
| | 0.9 | **0.0 [8.0 ± 0.0]** | 0.0 [8.0 ± 0.0] | 0.0 [8.0 ± 0.0] | 0.0 [8.0 ± 0.0] | 0.0 [8.0 ± 0.0] |
| Tree | 0.1 | **9.1 [8.0 ± 0.5]** | 9.7 [7.4 ± 0.9] | 10.9 [6.1 ± 0.4] | 12.4 [4.7 ± 0.4] | 11.4 [5.7 ± 0.5] |
| | 0.3 | **4.9 [12.8 ± 0.4]** | 5.0 [12.8 ± 0.6] | 6.0 [11.8 ± 0.4] | 5.2 [12.6 ± 0.4] | 7.4 [10.3 ± 0.5] |
| | 0.5 | 1.7 [16.1 ± 0.2] | **1.4 [16.4 ± 0.2]** | 2.4 [15.4 ± 0.3] | 1.4 [16.4 ± 0.2] | 4.5 [13.2 ± 0.4] |
| | 0.7 | 0.6 [17.2 ± 0.1] | **0.3 [17.4 ± 0.0]** | 0.6 [17.1 ± 0.2] | 0.6 [17.2 ± 0.1] | 2.5 [15.3 ± 0.3] |
| | 0.9 | 0.1 [17.7 ± 0.0] | **0.0 [17.7 ± 0.0]** | 0.1 [17.6 ± 0.2] | 0.2 [17.5 ± 0.1] | 0.1 [17.7 ± 0.1] |
| CliffWalk | 0.1 | 1152.9 [−1248.9 ± 117.3] | 520.8 [−616.8 ± 105.6] | 1166.1 [−1262.1 ± 119.2] | **296.1 [−392.0 ± 84.2]** | 1061.0 [−1157.0 ± 61.0] |
| | 0.3 | 285.9 [−370.0 ± 86.5] | 9.6 [−93.6 ± 0.4] | 378.5 [−462.5 ± 100.6] | **9.0 [−93.0 ± 0.4]** | 1190.5 [−1274.6 ± 118.9] |
| | 0.5 | 57.6 [−132.7 ± 32.8] | 14.3 [−89.4 ± 0.8] | 90.8 [−165.9 ± 46.2] | **11.7 [−86.7 ± 0.7]** | 1160.8 [−1235.8 ± 137.4] |
| | 0.7 | 32.7 [−98.9 ± 0.6] | 8.6 [−74.8 ± 2.2] | 33.8 [−100.0 ± 0.0] | **6.8 [−73.0 ± 1.9]** | 890.0 [−956.2 ± 136.6] |
| | 0.9 | 59.6 [−72.6 ± 3.9] | **25.6 [−38.6 ± 3.4]** | 87.0 [−100.0 ± 0.0] | 83.8 [−96.8 ± 1.5] | 412.8 [−425.8 ± 99.2] |
| TwoRooms | 0.1 | 1.0 [0.0 ± 0.0] | 1.0 [0.0 ± 0.0] | 1.0 [0.0 ± 0.0] | 1.0 [0.0 ± 0.0] | **0.7 [0.3 ± 0.0]** |
| | 0.3 | 0.9 [0.1 ± 0.0] | 0.9 [0.1 ± 0.0] | 0.9 [0.1 ± 0.0] | 0.9 [0.1 ± 0.0] | **0.5 [0.5 ± 0.1]** |
| | 0.5 | 0.8 [0.2 ± 0.0] | 0.8 [0.2 ± 0.0] | 0.8 [0.2 ± 0.0] | 0.8 [0.2 ± 0.0] | **0.3 [0.7 ± 0.0]** |
| | 0.7 | 0.7 [0.3 ± 0.0] | 0.6 [0.4 ± 0.0] | 0.5 [0.5 ± 0.1] | 0.5 [0.5 ± 0.1] | **0.1 [0.9 ± 0.0]** |
| | 0.9 | 0.3 [0.7 ± 0.0] | 0.2 [0.8 ± 0.0] | 0.1 [0.9 ± 0.0] | 0.2 [0.8 ± 0.0] | **0.0 [1.0 ± 0.0]** |

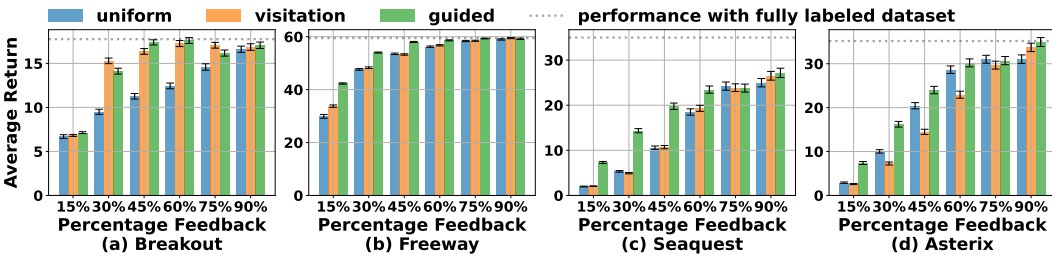

Figure 3: Comparison of `guided`, `visitation`, and `uniform` heuristic selection strategies on four large-scale domains: Breakout, Freeway, Seaquest, and Asterix. For each domain, the plot shows the mean policy return with error bars indicating the standard error.

total number of unique states $|\mathcal{S}|$ in each dataset, i.e., **Percentage Feedback $= B/|\mathcal{S}|$**, allowing for consistent comparison across domains.

### 4.1 PERFORMANCE OF HEURISTIC REWARD SELECTION DEPENDS ON DOMAINS TRAITS

We observe that the effectiveness of heuristic-based strategies is highly domain-dependent, and no single strategy consistently dominates. The results on prototypical domains are presented in Table 1 and on large-scale domains Figure 3. Additional experiments are deferred to Appendix D.3. The experiments for prototypical domains are averaged over 100 seeds, while results for large-scale domains are averaged over 10 seeds. Below, we highlight several key empirical findings:

1. **At low budgets:** When the reward labeling budget is small, the Q-values estimated form partially reward-labeled data are largely inaccurate. In such cases, `visitation` sampling generally provides an effective auxiliary signal for state selection. For example, in Graph, the visitation distribution induced by the data-collecting policy aligns well with the optimal selection even at low budgets, leading to improved performance. In CliffWalk, however, the on-policy visitation distribution proves to be more effective, as shown in Table 1.

2. **At high budgets:** As the budget increases, the learned Q-function becomes more accurate and informative. The *exploit*-term of `guided` sampling (see Equation 4) which relies on these Q-values to discover high-value states tends to aid performance and the `guided` strategy generally performes well. This trend is observed in 80% of the domains studied. Appendix D.8 outlines how the decay schedule and related parameters shape the exploration–exploitation tradeoff, and Appendix D.7 shows the strategy remains effective even with random initial samples.

3. **Impact of bottleneck structures:** In domains with bottleneck states—states that are chokepoints between regions of the environment, such as in TwoRooms and FrozenLake—sampling based on the visitation distribution under the data-collecting policy $\pi_D$ may overlook these infrequently visited but critical states. The bottleneck states need to be reward-labeled early on to facilite effective policy learning as the budget grows. In such cases, `uniform` sampling has a higher likelihood of sampling these states and tends to outperforms other heuristics by providing broader coverage across the entire state set.

**Takeaway:** The `guided` sampling strategy balances the strengths of `visitation` sampling at low budgets with those of sampling near high-value states at high budgets, making it a useful heuristic. However, in general, training-free heuristic-based reward selection must rely on discernible domain traits (further elaborated in Section 4.3) and the available labeling budget. Rather than a single heuristic being universally effective, performance is maximized by combining different heuristics—adapting them to budget levels (e.g., `guided`) and to domain-specific characteristics.

### 4.2 TRAINING PHASE FACILITATES NEAR-OPTIMAL PERFORMANCE

The strategies optimized using the training phase have higher performance than training-free strategies, with performance increasing along with the training cost of the strategy, as summarized in Figure 4. The `brute-force` strategy, while being optimal, has a prohibitive training cost for even small state sets.

For example, in a domain with $|\mathcal{S}| = 50$, exhaustively evaluating all possible state sets of size $B = 25$ would require $\binom{50}{25} \approx 10^{14}$ calls to the evaluator. Even at a rate of 2,000 calls to the evaluator per minute, completing this search would take about one hundred thousand years. The right y-axis in Figure 4 shows the estimated runtime based on a similar analysis. We compare this strategy with others on prototypical domains, in Figure 5. A reduced version of the strategy is studied on the MinAtar domains in Appendix D.4.

The training cost of `sequential-greedy` scales linearly with the size of the state set and the budget $(O(B|\mathcal{S}|))$, while the training cost of `ES` is independent of the state set and the budget and is determined only by the population size per iteration $m$ and number of iterations $k$. `ES` with cost $O(km)$ is denoted as `ES km`. On prototypical domains, we set $k = 10$

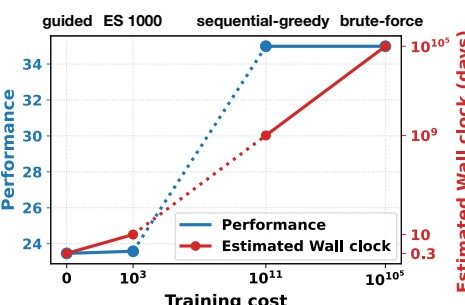

Figure 4: Performance vs. training cost for selection strategies in Seaquest (60% feedback). Optimal strategies require prohibitive training cost (right), while cost-efficient and heuristic approaches trade off some performance (left). The dotted region indicates where cost-efficient strategies could emerge.

and evaluate two variants: `ES 50` ($m = 5$) and `ES 200` ($m = 20$). Additional ablations of $k$ and $m$ are provided in Appendix D.4, where for large-scale domains, we run `ES 1000` to accommodate the greater size of the state space. The results in Figure 5 and Appendix D.4 yield the following key findings:

- **`sequential-greedy` is near-optimal** with significantly lower training cost than `brute-force`, establishing that training cost *linear* in $|\mathcal{S}|$ suffices for near-optimal performance. Greedy maximization of marginal utility (Equation 5) proves to be rather effective.
- **Zero-cost `guided` is comparable with low-cost `ES`** (Figure 5, Table 8), indicating that training-free guided sampling is preferable over low training costs, while `ES` is advantageous when performance can be scaled through higher $k$ and $m$.
- Optimal state sets identified on training datasets generalize well to test datasets from different data-collecting policies (Section 2), indicating **robustness to moderate dataset distribution shifts**. This is because for Q-learning based policy (as in `RLLF`), the occurrence of a state matters more than its frequency of occurrence, which may differ across datasets. This is further supported by Appendix D.6, where datasets collected under varying behavior policies yield consistent results.

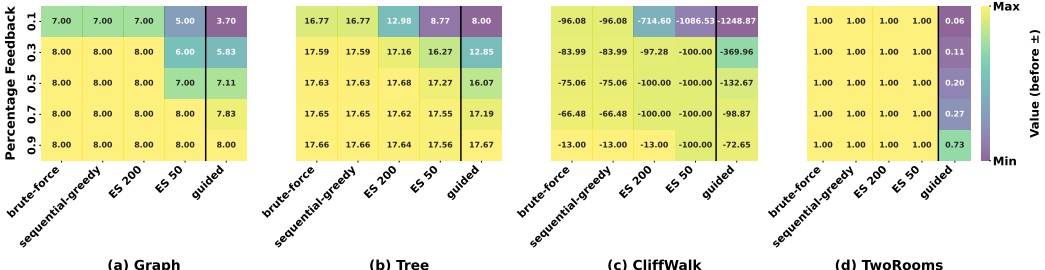

Figure 5: Performance comparison of training-phase strategies and training-free `guided` on prototypical domains. Values show mean policy return over five test datasets (standard errors negligible). `sequential-greedy` achieves near-optimal performance, while `guided` is comparable to `ES`.

### 4.3 COMMON STRUCTURAL PATTERNS OF OPTIMAL REWARD SELECTIONS

As noted earlier, effective selection methods yield near-optimal policies with significantly fewer reward labels than full supervision. We now conduct a post-hoc examination of the optimal state sets $(\mathcal{S}^*_{[B]})$ across domains to identify recurring structural patterns.

**Pattern 1: Prioritizing Optimal Pathways.** Optimal state sets include states that serve as anchor points to keep the agent on high-return paths in the domain. This is particularly evident in deterministic domains (like Graph), where at low budgets the goal state is selected first, after which $\mathcal{S}^*_{[B]}$ expands

as the budget increases to include additional anchor points. In sparse reward domains, particularly those without hazardous or penalty states, optimal selection depends mainly on identifying states with high rewards: an example being paddle-ball alignment in Breakout as shown in Appendix D.5.

**Pattern 2: Coverage of Near-Optimal Paths.** Particulary for domains with stochastic transitions (like Tree), states in $\mathcal{S}^*_{[B]}$, i.e., states that get reward-labeled with high priority, include those that lie in the vicinity of optimal pathways of the domain. They serve to facilitate recovery back onto the optimal pathways from deviations that may occur due to stochastic transitions.

**Pattern 3: Early Labeling of Penalty States.** Penalty states (such as terminal or trap states in FrozenLake and TwoRooms-Trap) get reward-labeled early on, even at low budgets. This ensures that subsequently learned policies steer away from these states, serving the role opposite of anchor points.

**Takeaway:** These patterns indicate that optimal state sets follow intuitive structural roles: anchoring trajectories on high-return paths, supporting recovery in stochastic settings, and steering policies away from hazards. Overall, it matters that pathways along with reward information flows get selected. When such information is available a priori—through expert demonstrations or domain knowledge—design for effective reward selection strategies must explicitly emphasize these traits, in addition to the broader considerations of domain properties and budget levels discussed earlier for heuristic approaches.

## 5 RELATED WORK

The problem of reward selection for RLLF remains largely unexplored. The closest formalization is by Parisi et al. (2024b), who consider *partially observable rewards* in online RL, but their setting conflates exploration with reward acquisition, making the focus different from our purely offline formulation. Zhan et al. (2023) propose a sampling approach for reward annotation but assume linear reward models, whereas our method does not impose such structural constraints. *Active RL* studies querying strategies under online exploration constraints, where agents must pay to observe rewards (Krueger et al., 2020; Schulze & Evans, 2018; Tucker et al., 2023). Our setting differs fundamentally: we study offline data with no additional exploration burden. Relatedly, Konyushova et al. (2021) address active off-policy data selection to improve policy evaluation, focusing on policy-level data collection rather than fine-grained reward state selection.

Works on *active reward learning* (Sadigh et al., 2017; Bıyık et al., 2019; Wilde et al., 2020; Daniel et al., 2015; Lindner et al., 2021) study how to query feedback that improves the generalization of a learned reward function. Our formulation differs fundamentally: because individual rewards are not retained, no reward model can be learned, and the focus shifts to improving policy learning rather than reward estimation. Other recent work explores reward modeling under uncertainty, for example, using priors over reward functions (Hu et al., 2023) or studying data influence (Munos & Moore, 2002; Koh & Liang, 2017; Gottesman et al., 2020). We complement these analyses by studying how selectively adding reward labels to previously unlabeled data influences the resulting policy performance.

The use of non reward labeled data has been studied for *online (state-based) exploration with unlabeled samples*. Some methods pseudo-label unlabeled samples to improve online exploration (Wilcoxson et al., 2024; Li et al., 2024), or develop exploration algorithms that operate under missing reward labels (Parisi et al., 2024a; Huang et al.). However, these primarily study exploration dynamics, whereas our focus is purely on optimizing offline reward label acquisition. A detailed comparison with these and additional works is provided in Appendix B.

## 6 DISCUSSION AND CONCLUSION

We introduce reward selection as a critical but underexplored challenge in RLLF. By decoupling selection from policy learning, we present the first systematic evaluation of zero-shot heuristics and optimized strategies across diverse environments, defining simple yet strong baselines and offering insights for future reward-efficient algorithms in domains like RLHF and drug discovery. The effectiveness of reward selection varies with domain dynamics and reward structure: in deterministic settings with frequent rewards, path-following heuristics perform well; in stochastic or sparse-reward domains,strategies that promote broader state coverage prove more effective. No single heuristic

dominates across all cases, and effective selection must align with both the domain and learning algorithm. Our findings establish reward selection as a powerful paradigm for scaling reinforcement learning in limited feedback settings.

While our study focuses on value-based policy updates, extending selection strategies to policy-gradient methods is a promising direction. Additionally, our general framework abstracts away domain-specific structure; however, incorporating inductive biases, such as temporal correlations in time-series tasks, may further aid selection strategies. Exploring how to integrate such structured priors offers an exciting path for future work.

**Reproducibility Statement:** We provide detailed description of each selection strategy in Section 3, with additional details about hyperparameters and domains in Appendix D. Code is added as supplementary material to the submission and will be made public upon acceptance.

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

## A    ADDITIONAL MOTIVATING EXAMPLES

1. **Reinforcement Learning from Human Feedback (RLHF) in LLMs:** In training large language models (LLMs), model-generated outputs are plentiful, but high-quality human preference labels remain costly and scarce (Ouyang et al., 2022; Christiano et al., 2017). This creates a reward selection challenge: which model completions should be labeled with human feedback to best guide downstream policy improvement? This mirrors our setup, where a budgeted selection of feedback points must be made to train a performant policy while minimizing labeling operational cost (ABAKA AI, 2025).

2. **AI-driven Drug Discovery:** Generative models can propose vast libraries of candidate molecules (Gómez-Bombarelli et al., 2018; Reymond, 2015; Jin et al., 2019), but only a limited subset can be experimentally evaluated for synthesizability, bioactivity, and toxicity due to the cost and time of wet-lab trials (DiMasi et al., 2016). Reward selection here involves choosing which molecular candidates to evaluate, analogous to selecting states for reward labeling in our framework to maximize downstream performance within a practically limited evaluation budget.

3. **Autonomous Driving:** Simulation platforms can produce diverse driving trajectories across environments and policies at scale (Dosovitskiy et al., 2017), but obtaining expert evaluations—such as comfort, rule compliance, or safety—is resource-intensive (Feng et al., 2023). Thus, a reward selection strategy is needed to determine which trajectories to annotate to yield robust, deployable policies, much like our proposed approach to feedback-efficient learning.

4. **Robotics:** Simulated environments enable generation of numerous trajectories, but transferring and evaluating those policies in the real world involves expensive and time-consuming physical experiments (Rajeswaran et al., 2017; Chebotar et al., 2019). Reward selection in this domain involves prioritizing which simulated or real-world interactions to evaluate, paralleling our method's goal of selecting the most informative reward-labeled samples for efficient policy learning under cost constraints.

## B    EXTENDED RELATED WORK

The setup of active reward selection for `RLLF` has not been previously explored much. The closest formulation of this problem is in Parisi et al. (2024b), who provide a formulation for *partially observable rewards* in online RL and propose algorithms for that setting. The online formulation conflates the difficulty on online exploration with the utility of rewards, the latter being the focus of this work. sampling approach to acquiring exploratory trajectories that enable accurate learning of hidden reward functions before collecting any human feedback. Zhan et al. (2023) propose a sampling approach to acquire data to be reward-annotated, although their analysis assumes linearity of reward functions. Similar to discovering high-utility reward states, Konyushova et al. (2021) study active collection of online data to determine promising policies and improve their performance estimates, as active off-policy selection.

The topic of reward selection has been studied under *Active RL*, which is perhaps closest in its motivation to our setting: where the agent must pay a cost to observe the reward, although for an online setting, yet again conflating the difficulty of exploration with the utility of rewards. Krueger et al. (2020) study this in the bandit setting, while Tucker et al. (2023) extend it to structured settings but retain the bandit-style objective of identifying the best arm by using reward queries to increase confidence in the average (stochastic) outcomes of each arm. This differs from our problem in two major ways: the stochasticity of rewards for each arm forces repeated sampling, and the lack of sequentiality of actions (leading to different outcomes for repeated pulls of the same arm) shifts the focus from reward utility to uncertainty mitigation. In contrast, Schulze & Evans (2018) propose a Bayes-optimal algorithm using Monte Carlo Tree Search (MCTS) to actively select reward observations. Finally, approaches like Lindner et al. (2021) actively select queries to maximize information gain about the reward function for modeling it.

The use of non-reward-labeled data has been extensively explored in the context of *online state-based exploration with unlabeled samples*. Wilcoxson et al. (2024) propose assigning pseudolabels to unlabeled data to guide exploration, while Li et al. (2024) leverage prior offline datasets and online rewards to pseudo-label new data for improved exploration. Parisi et al. (2024a) examine exploration under partially observed rewards, a setting closely related to ours but focused on online interaction. Huang et al. introduce a data collection strategy combining online RL with offline datasets to approach the performance of the optimal policy. Yu et al. (2022) show that setting unknown rewards to zero can perform surprisingly well in certain offline RL settings, a finding we also confirm in our experiments. Hu et al. (2023) propose using unlabeled data by assuming priors over possible reward functions and optimizing over sampled realizations of those reward functions.

Beyond data-driven exploration, *influence functions* have been proposed as signals for high-utility rewards. Munos & Moore (2002) defines the influence of a reward on value as $\frac{\partial V^*(s)}{\partial R(s')}$, equivalent to the state visitation frequency under the optimal policy. Other works, such as Koh & Liang (2017) and Gottesman et al. (2020), analyze the effect of removing known datapoints on prediction performance. In contrast, we study the anticipated influence of adding partially unknown datapoints, requiring assumptions about their potential impact. Finally, Lindner et al. (2021) provide an algorithm for learning reward models independently of the reward querying process, which relates directly to the focus of our study.

## C  ADDITIONAL NOTES ON METHODOLOGY

### C.1  CATEGORIZATION OF REWARD SELECTION STRATEGIES INVESTIGATED

We categorize the reward state selection strategies introduced in Section 3 according to three key design dimensions: (i) whether selection during the test phase is performed in an *open-loop* or *closed-loop* manner, (ii) whether training-phase selection operates in a *batch* or *iterative* mode, and (iii) the degree to which each strategy utilizes the *evaluator* during training. Table 2 presents a high-level taxonomy across these dimensions.

| Selection Strategy | Test: Open/Closed Loop | Train: Batch/Iterative | Train: Evaluator Use |
|---|---|---|---|
| **Trained Strategies** | | | |
| `brute-force` | Open loop | Batch | Yes |
| `sequential-greedy` | Open loop | Iterative | Yes |
| `evolutionary-strategy` | Open loop | Batch | Yes |
| **Training-free Heuristics** | | | |
| `guided` | Closed loop | Iterative | No |
| `guided-on-policy` | Closed loop | Iterative | No |
| `visitation` | Open loop | Batch | No |
| `visitation-on-policy` | Closed loop | Iterative | No |
| `uniform` | Open loop | Batch | No |

Table 2: Categorization of reward selection methods by design dimensions. Columns are shaded to distinguish test-phase (green) and training-phase (blue) attributes. Methods are grouped based on whether they use the evaluator during training.

### C.2  DESCRIPTION AND NOTATION FOR ITERATIVE REWARD SELECTION STRATEGIES

Iterative reward selection strategies construct the reward-labeled state set $\mathcal{S}_{[B]}$ in a sequential manner. At each step $b \in \{1, \ldots, B\}$, a new state $s_b \in \mathcal{S}$ is selected—conditioned on relevant information such as the current estimates of the Q-values of the policy or current policy's state-visitation distribution—and added to the selection set $\mathcal{S}_{[b-1]}$ to form $\mathcal{S}_{[B]}$. Relevant notation:

- $\mathcal{S}_{[b]}$: The set of selected states after $b$ iterations, i.e., $\mathcal{S}_{[b]} = \mathcal{S}_{[b-1]} \cup \{s_b\}$.

- $q_b$: The selection strategy or distribution used to sample the next state $s_b$ at iteration $b$, potentially conditioned on policy information or prior selections.

- $\pi_{[b]}$: The intermediate policy obtained after the $b^{\text{th}}$ reward selection and updated via RLLF.

- $Q^{\pi_{[b-1]}}$: The Q-function corresponding to $\pi_{[b-1]}$ after the $(b-1)^{\text{th}}$ reward selection and update.

# D  ADDITIONAL EXPERIMENTS AND EMPIRICAL DETAILS

## D.1  DOMAIN DETAILS

Table 3 summarizes the domains and their corresponding experimental setup. We study six prototypical domains (Graph, Tree, TwoRooms, TwoRooms-Trap, FrozenLake, and CliffWalk) and four large-scale MinAtar domains (Breakout, Freeway, Seaquest, and Asterix). The Graph, Tree, TwoRooms, and TwoRooms-Trap domains are custom-designed to expose structural properties relevant for analyzing reward selection strategies, while FrozenLake and CliffWalk are standard Gymnasium benchmarks (Brockman et al., 2016; Foundation, 2023).

Table 3: Summary of domains and their experimental setup.

|  | Prototypical Domains | Large-scale Domains (MinAtar) |
|---|---|---|
| Domain Names | Graph, Tree, TwoRooms, TwoRooms-Trap, FrozenLake, CliffWalk | Breakout, Freeway, Seaquest, Asterix |
| State Representation | Numeric (tabular) | Image-based ($10\times10$ pixels) |
| Expert Policy | Value Iteration | Online DQN |
| Policy Learning Algorithm | Offline Q-learning | Implicit Q-learning (IQL) |

**Domain description**  Brief descriptions of all domains are provided below.

- **Graph**: A two-row graph structure with 8 nodes per row. In each adjacent column, the $2 \times 2$ nodes are fully connected. Transitions are deterministic; actions move the agent between rows or advance to the next column in the same row. States correspond to nodes; every movement yields a dense reward.

- **Tree**: A complete binary tree where actions correspond to moving left or right. Transitions are stochastic: the agent moves in the intended direction with 85% probability and in the alternate direction with 15%. Rewards are dense and provided at every step.

- **TwoRooms**: Two $5 \times 5$ gridworld rooms connected by a narrow bottleneck state. The agent starts in one room and must reach a goal located in the other. Rewards are sparse: zero everywhere except a reward of $1$ at the goal state.

- **TwoRooms-Trap**: A variant of TwoRooms with six additional trap states. Entering a trap terminates the episode immediately with a penalty of $-100$. The environment otherwise shares the layout and reward structure of TwoRooms.

- **FrozenLake**: A standard Gymnasium benchmark (Brockman et al., 2016; Foundation, 2023). The agent navigates a slippery grid from start to goal, avoiding holes that cause termination. Transitions are stochastic and rewards are sparse (reward only at the goal).

- **CliffWalk**: Another Gymnasium benchmark. The agent must traverse a grid from start to goal while avoiding a high-penalty cliff region. Transitions are deterministic.

- **Minatar**: A set of simplified Atari-inspired environments with compact state and action spaces (Young & Tian, 2019). We evaluate on Breakout, Freeway, Seaquest, and Asterix.

**Policy training**  For prototypical domains, expert policies are generated using value iteration and policies are trained with offline Q-learning. For large-scale MinAtar domains, expert policies are obtained by training online DQN agents, and offline learning uses implicit Q-learning (IQL). Prototypical domains use tabular Q-functions due to their discrete, low-dimensional state spaces, while large-scale domains rely on neural network approximators for Q-values, given their high-dimensional $10 \times 10$ image-based states.

**Dataset collection**  Datasets are collected using a mixture-based data-collecting policy that combines expert and random actions. At each timestep, the agent follows the expert policy with probability $\epsilon$ and takes a uniformly random action with probability $1 - \epsilon$. For training, we use a single data-collecting policy with $\epsilon = 0.5$. For evaluation, five test data-collecting policies are created with $\epsilon \in \{0.55, 0.53, 0.51, 0.48, 0.45\}$ to study the robustness of learned policies under small distribution shifts.

**Compute resources**   All experiments on prototypical domains were conducted on CPUs, while those on large-scale domains were run on GeForce RTX 2080 Tis.

### D.2   DIFFERENT REWARD-LABELED-SETS RESULT IN POLICIES WITH VARYING PERFORMANCE

In Figure 1, we illustrate that different reward-labeled sets lead to policies with varying performance. We empirically validate this observation in two prototypical domains, Graph and Tree. For each domain, we select three percentage feedbacks (20%, 40%, and 60%), and report the average return of policies learned from all possible combinations at that budget. For example, in the Graph domain, which has 16 total states, selecting $b = 2$ yields $\binom{16}{2} = 120$ possible combinations; we report the average return across policies trained on datasets labeled by each of these 120 state sets. The results, shown in Figure 6, demonstrate that for a fixed budget, different combinations of labeled states can lead to significantly different policy performance.

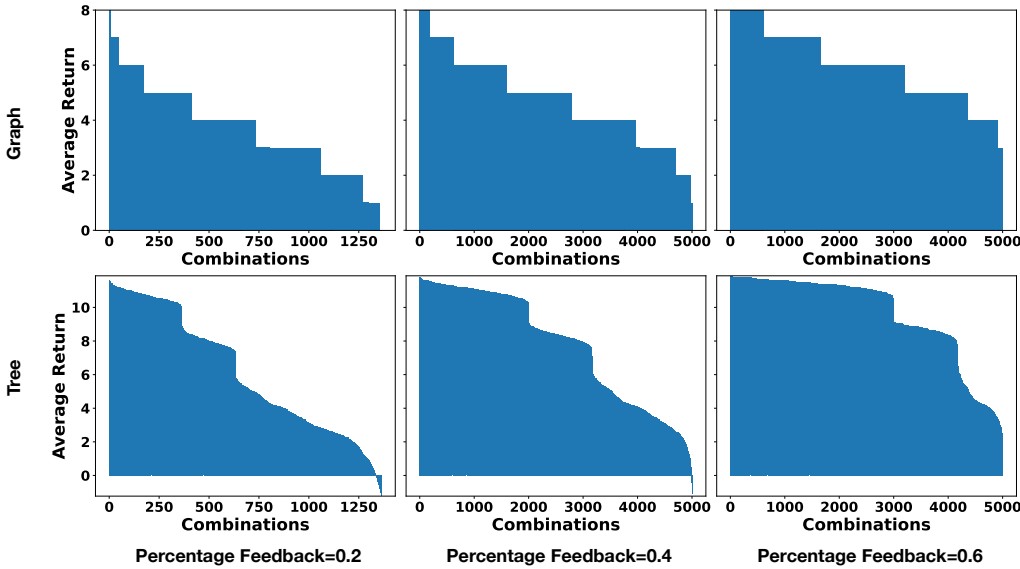

Figure 6: Performance variability across different reward-labeled state sets at fixed budgets. The first row shows results for the Graph domain; the second row shows results for the Tree domain. Columns correspond to percentage feedback levels of 20%, 40%, and 60%, respectively. The results illustrate that at the same feedback level, the choice of which states are labeled strongly affects the resulting policy performance.

### D.3 ADDITIONAL RESULTS FOR HEURISTIC-BASED SELECTION

The heuristics results on all prototypical domains are shown in Table 4, which aligns with the findings we've got in Section 4.1. Results on FrozenLake and TwoRooms-Trap domains have a similar pattern to TwoRooms domain, as their reward function are all sparse, and have bottleneck states.

Table 4: Comparison of `guided`, `visitation`, and `uniform` heuristic selection strategies on prototypical domains. For each domain, the table presents the mean policy return ($\pm$ standard error) and the corresponding optimality gap (in parentheses) across five percentage feedback levels.

| Domains | Percentage Feedback | guided | guided-on-policy | visitation | visitation-on-policy | uniform |
|---|---|---|---|---|---|---|
| Graph | 0.1 | $3.701 \pm 0.129$ (3.302) | $3.208 \pm 0.139$ (3.795) | $\mathbf{3.797 \pm 0.151}$ (**3.206**) | $3.300 \pm 0.142$ (3.703) | $2.949 \pm 0.137$ (4.054) |
| | 0.3 | $5.831 \pm 0.137$ (2.169) | $5.760 \pm 0.127$ (2.240) | $\mathbf{5.871 \pm 0.146}$ (**2.129**) | $5.690 \pm 0.146$ (2.310) | $4.617 \pm 0.156$ (3.383) |
| | 0.5 | $7.110 \pm 0.099$ (0.890) | $\mathbf{7.690 \pm 0.070}$ (**0.310**) | $7.199 \pm 0.090$ (0.801) | $7.583 \pm 0.086$ (0.417) | $5.978 \pm 0.114$ (2.022) |
| | 0.7 | $7.830 \pm 0.040$ (0.170) | $\mathbf{8.000 \pm 0.000}$ (**0.000**) | $7.599 \pm 0.060$ (0.401) | $7.991 \pm 0.009$ (0.009) | $6.920 \pm 0.084$ (1.080) |
| | 0.9 | $\mathbf{8.000 \pm 0.000}$ (**0.000**) | $\mathbf{8.000 \pm 0.000}$ (**0.000**) | $\mathbf{8.000 \pm 0.000}$ (**0.000**) | $\mathbf{8.000 \pm 0.000}$ (**0.000**) | $\mathbf{8.000 \pm 0.000}$ (**0.000**) |
| Tree | 0.1 | $\mathbf{8.003 \pm 0.468}$ (**9.053**) | $7.403 \pm 0.869$ (9.653) | $6.133 \pm 0.428$ (10.924) | $4.658 \pm 0.370$ (12.398) | $5.665 \pm 0.532$ (11.392) |
| | 0.3 | $\mathbf{12.846 \pm 0.373}$ (**4.921**) | $12.755 \pm 0.632$ (5.013) | $11.763 \pm 0.427$ (6.004) | $12.601 \pm 0.414$ (5.167) | $10.341 \pm 0.524$ (7.427) |
| | 0.5 | $16.072 \pm 0.205$ (1.695) | $\mathbf{16.415 \pm 0.207}$ (**1.352**) | $15.395 \pm 0.297$ (2.372) | $16.379 \pm 0.216$ (1.388) | $13.218 \pm 0.430$ (4.550) |
| | 0.7 | $17.193 \pm 0.083$ (0.575) | $\mathbf{17.444 \pm 0.037}$ (**0.323**) | $17.135 \pm 0.153$ (0.633) | $17.174 \pm 0.120$ (0.594) | $15.258 \pm 0.312$ (2.509) |
| | 0.9 | $17.673 \pm 0.013$ (0.094) | $\mathbf{17.731 \pm 0.031}$ (**0.036**) | $17.609 \pm 0.158$ (0.049) | $17.521 \pm 0.110$ (0.246) | $17.695 \pm 0.141$ (0.072) |
| CliffWalk | 0.1 | $-1248.872 \pm 117.272$ (1152.914) | $-616.760 \pm 105.578$ (520.803) | $-1262.067 \pm 119.207$ (1166.109) | $\mathbf{-392.040 \pm 84.184}$ (**296.082**) | $-1156.960 \pm 61.025$ (1061.002) |
| | 0.3 | $-369.964 \pm 86.539$ (285.948) | $-93.637 \pm 0.373$ (9.621) | $-462.530 \pm 100.633$ (378.515) | $\mathbf{-92.981 \pm 0.358}$ (**8.965**) | $-1274.561 \pm 118.910$ (1190.545) |
| | 0.5 | $-132.671 \pm 32.819$ (57.629) | $-89.390 \pm 0.827$ (14.348) | $-165.870 \pm 46.201$ (90.828) | $\mathbf{-86.746 \pm 0.677}$ (**11.704**) | $-1235.823 \pm 137.366$ (1160.781) |
| | 0.7 | $-98.870 \pm 0.647$ (32.665) | $-74.821 \pm 2.171$ (8.615) | $-100.000 \pm 0.000$ (33.794) | $\mathbf{-72.995 \pm 1.872}$ (**6.790**) | $-956.208 \pm 136.611$ (890.003) |
| | 0.9 | $-72.646 \pm 3.909$ (59.646) | $\mathbf{-38.592 \pm 3.373}$ (**25.592**) | $-100.000 \pm 0.000$ (87.000) | $-96.819 \pm 1.466$ (83.819) | $-425.837 \pm 99.188$ (412.837) |
| FrozenLake | 0.1 | $0.021 \pm 0.007$ (−0.721) | $0.056 \pm 0.017$ (−0.686) | $0.028 \pm 0.010$ (−0.714) | $0.020 \pm 0.007$ (−0.722) | $\mathbf{0.145 \pm 0.028}$ (**−0.598**) |
| | 0.3 | $0.087 \pm 0.022$ (−0.655) | $0.078 \pm 0.021$ (−0.663) | $0.079 \pm 0.021$ (−0.663) | $0.050 \pm 0.016$ (−0.692) | $\mathbf{0.306 \pm 0.036}$ (**−0.436**) |
| | 0.5 | $0.165 \pm 0.029$ (−0.578) | $0.127 \pm 0.026$ (−0.617) | $0.171 \pm 0.030$ (−0.573) | $0.086 \pm 0.022$ (−0.657) | $\mathbf{0.467 \pm 0.036}$ (**−0.276**) |
| | 0.7 | $0.261 \pm 0.034$ (−0.482) | $0.251 \pm 0.034$ (−0.492) | $0.326 \pm 0.036$ (−0.416) | $0.160 \pm 0.029$ (−0.582) | $\mathbf{0.582 \pm 0.031}$ (**−0.160**) |
| | 0.9 | $0.477 \pm 0.035$ (−0.263) | $0.508 \pm 0.033$ (−0.232) | $0.566 \pm 0.031$ (−0.174) | $0.427 \pm 0.036$ (−0.313) | $\mathbf{0.697 \pm 0.019}$ (**−0.043**) |
| TwoRooms | 0.1 | $0.012 \pm 0.010$ (0.988) | $0.022 \pm 0.014$ (0.978) | $0.042 \pm 0.020$ (0.959) | $0.022 \pm 0.014$ (0.979) | $\mathbf{0.261 \pm 0.044}$ (**0.739**) |
| | 0.3 | $0.077 \pm 0.027$ (0.923) | $0.092 \pm 0.029$ (0.908) | $0.071 \pm 0.025$ (0.929) | $0.081 \pm 0.027$ (0.919) | $\mathbf{0.530 \pm 0.050}$ (**0.470**) |
| | 0.5 | $0.173 \pm 0.039$ (0.827) | $0.151 \pm 0.036$ (0.849) | $0.182 \pm 0.038$ (0.818) | $0.181 \pm 0.038$ (0.819) | $\mathbf{0.720 \pm 0.045}$ (**0.280**) |
| | 0.7 | $0.270 \pm 0.046$ (0.730) | $0.371 \pm 0.048$ (0.629) | $0.481 \pm 0.050$ (0.519) | $0.501 \pm 0.050$ (0.499) | $\mathbf{0.910 \pm 0.029}$ (**0.090**) |
| | 0.9 | $0.732 \pm 0.046$ (0.268) | $0.800 \pm 0.040$ (0.200) | $0.870 \pm 0.034$ (0.130) | $0.770 \pm 0.042$ (0.230) | $\mathbf{0.990 \pm 0.010}$ (**0.010**) |
| TwoRooms-Trap | 0.1 | $-58.492 \pm 0.642$ (59.492) | $-60.151 \pm 0.673$ (61.151) | $-59.390 \pm 1.156$ (60.390) | $-61.520 \pm 0.723$ (62.520) | $\mathbf{-46.850 \pm 2.884}$ (**47.850**) |
| | 0.3 | $-45.692 \pm 1.015$ (46.692) | $-47.391 \pm 0.899$ (48.391) | $-47.130 \pm 1.538$ (48.130) | $-49.960 \pm 1.022$ (50.960) | $\mathbf{-29.340 \pm 2.983}$ (**30.340**) |
| | 0.5 | $-16.440 \pm 0.968$ (17.440) | $-15.374 \pm 0.874$ (16.374) | $-23.320 \pm 1.396$ (24.320) | $-20.140 \pm 0.934$ (21.140) | $\mathbf{-13.270 \pm 2.261}$ (**14.270**) |
| | 0.7 | $-0.336 \pm 0.056$ (1.336) | $\mathbf{-0.210 \pm 0.065}$ (**1.210**) | $-0.300 \pm 0.349$ (1.300) | $-0.700 \pm 0.160$ (1.700) | $-1.600 \pm 0.916$ (2.600) |
| | 0.9 | $\mathbf{1.000 \pm 0.000}$ (**0.000**) | $\mathbf{1.000 \pm 0.000}$ (**0.000**) | $\mathbf{1.000 \pm 0.000}$ (**0.000**) | $0.851 \pm 0.040$ (0.149) | $\mathbf{1.000 \pm 0.000}$ (**0.000**) |

### D.4 ADDITIONAL RESULTS FOR TRAINING-BASED STRATEGIES

In sparse-reward environments, `brute-force` search can be accelerated by recognizing that only states with non-zero rewards must be labeled. This greatly reduces the number of combinations to consider, making exact evaluation tractable in small domains. The optimality results on all prototypical domains are shown in Table 5, where we further show the error bar of each experiment, which are omitted in Section 4.2, as the results between different seeds are almost the same, showing the robustness of optimal selection strategies.

Table 5: Performance comparison of `brute-force`, `sequential-greedy`, and `ES` on prototypical domains. Results are reported on training datasets, with test performance shown in parentheses (e.g., train score (test score)). Test scores are reported as mean $\pm$ standard error across five test datasets. `ES 200` corresponds to $k = 10, m = 20$ and `ES 50` to $k = 10, m = 5$.

| Domains | Percentage Feedback | brute-force | sequential-greedy | ES 200 | ES 50 | guided |
|---|---|---|---|---|---|---|
| Graph | 0.1 | $7.003(7.003 \pm 0.000)$ | $7.003(7.003 \pm 0.000)$ | $7.003(7.003 \pm 0.000)$ | $4.999(4.996 \pm 0.000)$ | 3.701 |
| | 0.3 | $8.000(8.000 \pm 0.000)$ | $8.000(8.000 \pm 0.000)$ | $8.000(8.000 \pm 0.000)$ | $6.000(6.000 \pm 0.000)$ | 5.831 |
| | 0.5 | $8.000(8.000 \pm 0.000)$ | $8.000(8.000 \pm 0.000)$ | $8.000(8.000 \pm 0.000)$ | $7.003(7.003 \pm 0.000)$ | 7.110 |
| | 0.7 | $8.000(8.000 \pm 0.000)$ | $8.000(8.000 \pm 0.000)$ | $8.000(8.000 \pm 0.000)$ | $8.000(8.000 \pm 0.000)$ | 7.830 |
| | 0.9 | $8.000(8.000 \pm 0.000)$ | $8.000(8.000 \pm 0.000)$ | $8.000(8.000 \pm 0.000)$ | $8.000(8.000 \pm 0.000)$ | 8.000 |
| Tree | 0.1 | $17.056(16.773 \pm 0.000)$ | $17.056(16.773 \pm 0.000)$ | $12.990(12.978 \pm 0.000)$ | $8.841(8.768 \pm 0.000)$ | 8.003 |
| | 0.3 | $17.767(17.592 \pm 0.017)$ | $17.767(17.592 \pm 0.033)$ | $17.198(17.157 \pm 0.000)$ | $16.199(16.271 \pm 0.000)$ | 12.846 |
| | 0.5 | $17.767(17.629 \pm 0.020)$ | $17.767(17.629 \pm 0.018)$ | $17.781(17.680 \pm 0.000)$ | $17.445(17.275 \pm 0.009)$ | 16.072 |
| | 0.7 | $17.767(17.649 \pm 0.000)$ | $17.767(17.649 \pm 0.000)$ | $17.777(17.623 \pm 0.000)$ | $17.642(17.547 \pm 0.000)$ | 17.193 |
| | 0.9 | $17.767(17.657 \pm 0.000)$ | $17.767(17.657 \pm 0.000)$ | $17.736(17.639 \pm 0.000)$ | $17.746(17.564 \pm 0.000)$ | 17.673 |
| CliffWalk | 0.1 | $-95.958(-96.081 \pm 0.001)$ | $-95.958(-96.081 \pm 0.001)$ | $-713.261(-714.600 \pm 0.019)$ | $-1086.006(-1086.526 \pm 0.039)$ | $-1248.872$ |
| | 0.3 | $-84.016(-83.986 \pm 0.001)$ | $-84.016(-83.986 \pm 0.001)$ | $-97.237(-97.276 \pm 0.000)$ | $-100.000(-100.000 \pm 0.000)$ | $-369.964$ |
| | 0.5 | $-75.042(-75.059 \pm 0.001)$ | $-75.042(-75.059 \pm 0.001)$ | $-100.000(-100.000 \pm 0.000)$ | $-100.000(-100.000 \pm 0.000)$ | $-132.671$ |
| | 0.7 | $-66.206(-66.477 \pm 0.001)$ | $-66.206(-66.477 \pm 0.001)$ | $-100.000(-100.000 \pm 0.000)$ | $-100.000(-100.000 \pm 0.000)$ | $-98.870$ |
| | 0.9 | $-13.000(-13.000 \pm 0.000)$ | $-13.000(-13.000 \pm 0.000)$ | $-13.000(-13.000 \pm 0.000)$ | $-100.000(-100.000 \pm 0.000)$ | $-72.646$ |
| FrozenLake | 0.1 | $0.746(0.729 \pm 0.010)$ | $0.746(0.729 \pm 0.010)$ | $0.742(0.728 \pm 0.009)$ | $0.014(0.014 \pm 0.000)$ | 0.021 |
| | 0.3 | $0.746(0.736 \pm 0.006)$ | $0.746(0.736 \pm 0.006)$ | $0.738(0.702 \pm 0.010)$ | $0.738(0.730 \pm 0.008)$ | 0.087 |
| | 0.5 | $0.746(0.719 \pm 0.012)$ | $0.746(0.719 \pm 0.012)$ | $0.740(0.731 \pm 0.009)$ | $0.737(0.714 \pm 0.009)$ | 0.165 |
| | 0.7 | $0.746(0.728 \pm 0.007)$ | $0.746(0.728 \pm 0.007)$ | $0.733(0.730 \pm 0.010)$ | $0.742(0.737 \pm 0.002)$ | 0.261 |
| | 0.9 | $0.746(0.719 \pm 0.008)$ | $0.746(0.719 \pm 0.008)$ | $0.739(0.740 \pm 0.001)$ | $0.743(0.734 \pm 0.005)$ | 0.477 |
| TwoRooms | 0.1 | $1.000(1.000 \pm 0.000)$ | $1.000(1.000 \pm 0.000)$ | $1.000(1.000 \pm 0.000)$ | $1.000(1.000 \pm 0.000)$ | 0.055 |
| | 0.3 | $1.000(1.000 \pm 0.000)$ | $1.000(1.000 \pm 0.000)$ | $1.000(1.000 \pm 0.000)$ | $1.000(1.000 \pm 0.000)$ | 0.109 |
| | 0.5 | $1.000(1.000 \pm 0.000)$ | $1.000(1.000 \pm 0.000)$ | $1.000(1.000 \pm 0.000)$ | $1.000(1.000 \pm 0.000)$ | 0.195 |
| | 0.7 | $1.000(1.000 \pm 0.000)$ | $1.000(1.000 \pm 0.000)$ | $1.000(1.000 \pm 0.000)$ | $1.000(1.000 \pm 0.000)$ | 0.270 |
| | 0.9 | $1.000(1.000 \pm 0.000)$ | $1.000(1.000 \pm 0.000)$ | $1.000(1.000 \pm 0.000)$ | $1.000(1.000 \pm 0.000)$ | 0.732 |
| TwoRooms-Trap | 0.1 | $1.000(1.000 \pm 0.000)$ | $1.000(1.000 \pm 0.000)$ | $1.000(1.000 \pm 0.000)$ | $1.000(1.000 \pm 0.000)$ | $-37.204$ |
| | 0.3 | $1.000(1.000 \pm 0.000)$ | $1.000(1.000 \pm 0.000)$ | $1.000(1.000 \pm 0.000)$ | $1.000(1.000 \pm 0.000)$ | $-16.440$ |
| | 0.5 | $1.000(1.000 \pm 0.000)$ | $1.000(1.000 \pm 0.000)$ | $1.000(1.000 \pm 0.000)$ | $1.000(1.000 \pm 0.000)$ | $-1.397$ |
| | 0.7 | $1.000(1.000 \pm 0.000)$ | $1.000(1.000 \pm 0.000)$ | $1.000(1.000 \pm 0.000)$ | $1.000(1.000 \pm 0.000)$ | 0.966 |
| | 0.9 | $1.000(1.000 \pm 0.000)$ | $1.000(1.000 \pm 0.000)$ | $1.000(1.000 \pm 0.000)$ | $1.000(1.000 \pm 0.000)$ | 1.000 |

In addition to the two `ES` variants presented in Section 4.2, we provide an ablation study examining how performance varies with different numbers of samples per iteration $m$ and iterations $k$. In Table 6, we fix $m = 20$ and vary $k$ across $\{3, 5, 8, 10\}$. In Table 7, we fix $k = 10$ and vary $m$ across $\{5, 10, 15, 20\}$. We find that larger values of $k \times m$ generally lead to better performance. Notably, increasing $m$ (the number of samples per iteration) tends to have a greater impact than increasing $k$ (the number of iterations), suggesting that sampling more candidates per iteration contributes more significantly to performance gains than simply running additional iterations.

Table 6: Ablation study of `ES` performance as a function of the number of iterations $k$ (with $m = 20$ fixed). Results are reported as $k \times m$ for consistency with the main paper (e.g., `ES` $10 \times 20$ indicates $k = 10$ and $m = 20$).

| Domains | Percentage Feedback | ES $10 \times 20$ | ES $8 \times 20$ | ES $5 \times 20$ | ES $3 \times 20$ |
|---|---|---|---|---|---|
| Graph | 0.1 | 7.003(7.003 ± 0.000) | 7.003(7.003 ± 0.000) | 7.003(7.003 ± 0.000) | 7.003(7.003 ± 0.000) |
| | 0.3 | 8.000(8.000 ± 0.000) | 8.000(8.000 ± 0.000) | 8.000(8.000 ± 0.000) | 8.000(8.000 ± 0.000) |
| | 0.5 | 8.000(8.000 ± 0.000) | 8.000(8.000 ± 0.000) | 8.000(8.000 ± 0.000) | 8.000(8.000 ± 0.000) |
| | 0.7 | 8.000(8.000 ± 0.000) | 8.000(8.000 ± 0.000) | 8.000(8.000 ± 0.000) | 8.000(8.000 ± 0.000) |
| | 0.9 | 8.000(8.000 ± 0.000) | 8.000(8.000 ± 0.000) | 8.000(8.000 ± 0.000) | 8.000(8.000 ± 0.000) |
| Tree | 0.1 | 12.990(12.978 ± 0.000) | 12.990(12.978 ± 0.000) | 12.880(12.884 ± 0.000) | 11.820(11.897 ± 0.086) |
| | 0.3 | 17.198(17.157 ± 0.000) | 17.329(17.111 ± 0.000) | 17.436(17.464 ± 0.000) | 16.357(16.161 ± 0.000) |
| | 0.5 | 17.781(17.680 ± 0.000) | 17.692(17.518 ± 0.009) | 17.583(17.535 ± 0.000) | 17.016(16.911 ± 0.000) |
| | 0.7 | 17.777(17.623 ± 0.000) | 17.763(17.603 ± 0.000) | 17.846(17.668 ± 0.000) | 17.721(17.552 ± 0.000) |
| | 0.9 | 17.736(17.639 ± 0.000) | 17.746(17.564 ± 0.000) | 17.746(17.564 ± 0.000) | 17.746(17.564 ± 0.000) |
| CliffWalk | 0.1 | −713.261(−714.600 ± 0.019) | −713.261(−714.567 ± 0.012) | −767.641(−769.536 ± 0.028) | −783.801(−786.146 ± 0.021) |
| | 0.3 | −97.237(−97.276 ± 0.000) | −97.329(−97.361 ± 0.000) | −95.920(−95.841 ± 0.001) | −100.000(−100.000 ± 0.000) |
| | 0.5 | −100.000(−100.000 ± 0.000) | −100.000(−100.000 ± 0.000) | −100.000(−100.000 ± 0.000) | −100.000(−100.000 ± 0.000) |
| | 0.7 | −100.000(−100.000 ± 0.000) | −100.000(−100.000 ± 0.000) | −100.000(−100.000 ± 0.000) | −100.000(−100.000 ± 0.000) |
| | 0.9 | −13.000(−13.000 ± 0.000) | −13.000(−13.000 ± 0.000) | −13.000(−13.000 ± 0.000) | −14.000(−13.996 ± 0.000) |
| FrozenLake | 0.1 | 0.742(0.728 ± 0.009) | 0.743(0.741 ± 0.001) | 0.740(0.740 ± 0.001) | 0.737(0.721 ± 0.010) |
| | 0.3 | 0.738(0.702 ± 0.010) | 0.740(0.727 ± 0.006) | 0.743(0.735 ± 0.006) | 0.738(0.735 ± 0.006) |
| | 0.5 | 0.740(0.731 ± 0.009) | 0.743(0.735 ± 0.005) | 0.740(0.710 ± 0.011) | 0.737(0.734 ± 0.005) |
| | 0.7 | 0.733(0.730 ± 0.010) | 0.740(0.714 ± 0.014) | 0.741(0.725 ± 0.013) | 0.739(0.710 ± 0.008) |
| | 0.9 | 0.739(0.740 ± 0.001) | 0.739(0.723 ± 0.007) | 0.742(0.710 ± 0.013) | 0.740(0.734 ± 0.005) |
| TwoRooms | 0.1 | 1.000(1.000 ± 0.000) | 1.000(1.000 ± 0.000) | 1.000(1.000 ± 0.000) | 1.000(1.000 ± 0.000) |
| | 0.3 | 1.000(1.000 ± 0.000) | 1.000(1.000 ± 0.000) | 1.000(1.000 ± 0.000) | 1.000(1.000 ± 0.000) |
| | 0.5 | 1.000(1.000 ± 0.000) | 1.000(1.000 ± 0.000) | 1.000(1.000 ± 0.000) | 1.000(1.000 ± 0.000) |
| | 0.7 | 1.000(1.000 ± 0.000) | 1.000(1.000 ± 0.000) | 1.000(1.000 ± 0.000) | 1.000(1.000 ± 0.000) |
| | 0.9 | 1.000(1.000 ± 0.000) | 1.000(1.000 ± 0.000) | 1.000(1.000 ± 0.000) | 1.000(1.000 ± 0.000) |
| TwoRooms-Trap | 0.1 | 1.000(1.000 ± 0.000) | 1.000(1.000 ± 0.000) | 1.000(1.000 ± 0.000) | 1.000(1.000 ± 0.000) |
| | 0.3 | 1.000(1.000 ± 0.000) | 1.000(1.000 ± 0.000) | 1.000(1.000 ± 0.000) | 1.000(1.000 ± 0.000) |
| | 0.5 | 1.000(1.000 ± 0.000) | 1.000(1.000 ± 0.000) | 1.000(1.000 ± 0.000) | 1.000(1.000 ± 0.000) |
| | 0.7 | 1.000(1.000 ± 0.000) | 1.000(1.000 ± 0.000) | 1.000(1.000 ± 0.000) | 1.000(1.000 ± 0.000) |
| | 0.9 | 1.000(1.000 ± 0.000) | 1.000(1.000 ± 0.000) | 1.000(1.000 ± 0.000) | 1.000(1.000 ± 0.000) |

Table 7: Ablation study of `ES` performance as a function of the number of samples per iteration $m$ (with $k = 10$ fixed). Results are reported as $k \times m$ for consistency with the main paper (e.g., `ES` $10 \times 20$ indicates $k = 10$ and $m = 20$).

| Domains | Percentage Feedback | ES $10 \times 20$ | ES $10 \times 15$ | ES $10 \times 10$ | ES $10 \times 5$ |
|---|---|---|---|---|---|
| Graph | 0.1 | 7.003(7.003 ± 0.000) | 5.999(6.001 ± 0.000) | 5.999(6.001 ± 0.000) | 4.999(4.996 ± 0.000) |
| | 0.3 | 8.000(8.000 ± 0.000) | 8.000(8.000 ± 0.000) | 8.000(8.000 ± 0.000) | 6.000(6.000 ± 0.000) |
| | 0.5 | 8.000(8.000 ± 0.000) | 8.000(8.000 ± 0.000) | 8.000(8.000 ± 0.000) | 7.003(7.003 ± 0.000) |
| | 0.7 | 8.000(8.000 ± 0.000) | 8.000(8.000 ± 0.000) | 8.000(8.000 ± 0.000) | 8.000(8.000 ± 0.000) |
| | 0.9 | 8.000(8.000 ± 0.000) | 8.000(8.000 ± 0.000) | 8.000(8.000 ± 0.000) | 8.000(8.000 ± 0.000) |
| Tree | 0.1 | 12.990(12.978 ± 0.000) | 12.754(12.301 ± 0.116) | 12.427(12.516 ± 0.000) | 8.841(8.768 ± 0.000) |
| | 0.3 | 17.198(17.157 ± 0.000) | 17.319(17.219 ± 0.000) | 17.082(17.015 ± 0.002) | 16.199(16.271 ± 0.000) |
| | 0.5 | 17.781(17.680 ± 0.000) | 17.454(17.334 ± 0.000) | 17.328(17.290 ± 0.034) | 17.445(17.275 ± 0.009) |
| | 0.7 | 17.777(17.623 ± 0.000) | 17.742(17.603 ± 0.000) | 17.727(17.726 ± 0.000) | 17.642(17.547 ± 0.000) |
| | 0.9 | 17.736(17.639 ± 0.000) | 17.736(17.639 ± 0.000) | 17.736(17.639 ± 0.000) | 17.746(17.564 ± 0.000) |
| CliffWalk | 0.1 | −713.261(−714.600 ± 0.019) | −755.425(−754.434 ± 0.000) | −867.262(−865.682 ± 0.021) | −1086.006(−1086.526 ± 0.039) |
| | 0.3 | −97.237(−97.276 ± 0.000) | −98.579(−98.576 ± 0.000) | −100.000(−100.000 ± 0.000) | −100.000(−100.000 ± 0.000) |
| | 0.5 | −100.000(−100.000 ± 0.000) | −100.000(−100.000 ± 0.000) | −100.000(−100.000 ± 0.000) | −100.000(−100.000 ± 0.000) |
| | 0.7 | −100.000(−100.000 ± 0.000) | −100.000(−100.000 ± 0.000) | −100.000(−100.000 ± 0.000) | −100.000(−100.000 ± 0.000) |
| | 0.9 | −13.000(−13.000 ± 0.000) | −13.000(−13.000 ± 0.000) | −100.000(−100.000 ± 0.000) | −100.000(−100.000 ± 0.000) |
| FrozenLake | 0.1 | 0.742(0.728 ± 0.009) | 0.739(0.698 ± 0.011) | 0.741(0.720 ± 0.013) | 0.014(0.014 ± 0.000) |
| | 0.3 | 0.738(0.702 ± 0.010) | 0.744(0.741 ± 0.002) | 0.740(0.728 ± 0.007) | 0.738(0.730 ± 0.008) |
| | 0.5 | 0.740(0.731 ± 0.009) | 0.739(0.723 ± 0.009) | 0.740(0.733 ± 0.005) | 0.737(0.714 ± 0.009) |
| | 0.7 | 0.733(0.730 ± 0.010) | 0.739(0.711 ± 0.014) | 0.739(0.722 ± 0.006) | 0.742(0.737 ± 0.002) |
| | 0.9 | 0.739(0.740 ± 0.001) | 0.738(0.737 ± 0.005) | 0.738(0.731 ± 0.008) | 0.743(0.734 ± 0.005) |
| TwoRooms | 0.1 | 1.000(1.000 ± 0.000) | 1.000(1.000 ± 0.000) | 1.000(1.000 ± 0.000) | 1.000(1.000 ± 0.000) |
| | 0.3 | 1.000(1.000 ± 0.000) | 1.000(1.000 ± 0.000) | 1.000(1.000 ± 0.000) | 1.000(1.000 ± 0.000) |
| | 0.5 | 1.000(1.000 ± 0.000) | 1.000(1.000 ± 0.000) | 1.000(1.000 ± 0.000) | 1.000(1.000 ± 0.000) |
| | 0.7 | 1.000(1.000 ± 0.000) | 1.000(1.000 ± 0.000) | 1.000(1.000 ± 0.000) | 1.000(1.000 ± 0.000) |
| | 0.9 | 1.000(1.000 ± 0.000) | 1.000(1.000 ± 0.000) | 1.000(1.000 ± 0.000) | 1.000(1.000 ± 0.000) |
| TwoRooms-Trap | 0.34 | 1.000(1.000 ± 0.000) | 1.000(1.000 ± 0.000) | 1.000(1.000 ± 0.000) | 1.000(1.000 ± 0.000) |
| | 0.48 | 1.000(1.000 ± 0.000) | 1.000(1.000 ± 0.000) | 1.000(1.000 ± 0.000) | 1.000(1.000 ± 0.000) |
| | 0.61 | 1.000(1.000 ± 0.000) | 1.000(1.000 ± 0.000) | 1.000(1.000 ± 0.000) | 1.000(1.000 ± 0.000) |
| | 0.75 | 1.000(1.000 ± 0.000) | 1.000(1.000 ± 0.000) | 1.000(1.000 ± 0.000) | 1.000(1.000 ± 0.000) |
| | 0.89 | 1.000(1.000 ± 0.000) | 1.000(1.000 ± 0.000) | 1.000(1.000 ± 0.000) | 1.000(1.000 ± 0.000) |

We also report `ES` results on large-scale MinAtar domains, using $k = 10, m = 100$ (`ES 1000`). Although the training computation of `ES` remains fixed, achieving accurate performance estimates still requires large $k \times m$ values. Even under this configuration, `ES` does not consistently outperform `guided`, illustrating the inherent difficulty of discovering optimal state sets in large state spaces even when an evaluator is available, as shown in Table 8.

In addition, Table 8 includes a column for reduced `brute-force`. By leveraging UDS, we only label the data points where rewards are non-zero. All four MinAtar domains exhibit sparse rewards, with fewer than 10% of states containing non-zero rewards. As a result, reduced `brute-force` is expected to identify a state set that achieves equivalent performance to the fully labeled dataset, while substantially reducing the labeling effort.

Table 8: Performance comparison of `ES` and `guided` for optimal state set selection on large-scale domains. Results are reported only on training datasets because the `guided` heuristic is defined with respect to the training dataset, and our comparison focuses on matching the settings for both methods. Although the training computation of `ES` is fixed, accurately evaluating its performance on large datasets remains costly, and small values of $k \times m$ yield poor results. Even with $k = 10, m = 100$ (denoted as `ES 1000`), `ES` does not consistently outperform `guided`. Scores are reported as mean $\pm$ standard error.

| Domains | Percentage Feedback | Reduced `brute-force` | `ES 1000` | `guided` |
|---|---|---|---|---|
| Breakout | 0.15 | | $17.75 \pm 0.85$ | $7.13 \pm 0.11$ |
| | 0.30 | | $17.66 \pm 0.40$ | $14.12 \pm 0.34$ |
| | 0.45 | 17.75 | $17.75 \pm 0.89$ | $17.39 \pm 0.29$ |
| | 0.60 | | $17.32 \pm 1.05$ | $17.60 \pm 0.33$ |
| | 0.75 | | $17.46 \pm 1.08$ | $16.17 \pm 0.35$ |
| | 0.90 | | $17.40 \pm 1.43$ | $17.06 \pm 0.37$ |
| Freeway | 0.15 | | $43.44 \pm 1.41$ | $42.31 \pm 0.25$ |
| | 0.30 | | $55.82 \pm 0.93$ | $54.01 \pm 0.21$ |
| | 0.45 | 58.28 | $58.28 \pm 0.48$ | $58.02 \pm 0.20$ |
| | 0.60 | | $58.28 \pm 0.46$ | $58.28 \pm 0.20$ |
| | 0.75 | | $58.28 \pm 0.81$ | $58.28 \pm 0.15$ |
| | 0.90 | | $58.28 \pm 0.45$ | $58.28 \pm 0.24$ |
| Seaquest | 0.15 | | $1.42 \pm 0.26$ | $7.30 \pm 0.23$ |
| | 0.30 | | $9.16 \pm 1.09$ | $14.35 \pm 0.47$ |
| | 0.45 | 34.99 | $18.80 \pm 2.25$ | $19.77 \pm 0.71$ |
| | 0.60 | | $23.58 \pm 3.00$ | $23.46 \pm 0.80$ |
| | 0.75 | | $24.99 \pm 3.44$ | $23.79 \pm 0.88$ |
| | 0.90 | | $25.48 \pm 3.31$ | $27.17 \pm 1.04$ |
| Asterix | 0.15 | | $4.88 \pm 0.74$ | $7.38 \pm 0.34$ |
| | 0.30 | | $9.06 \pm 1.10$ | $16.21 \pm 0.65$ |
| | 0.45 | 35.16 | $22.36 \pm 2.45$ | $24.00 \pm 0.84$ |
| | 0.60 | | $28.92 \pm 2.52$ | $30.19 \pm 0.91$ |
| | 0.75 | | $32.28 \pm 3.03$ | $30.71 \pm 0.94$ |
| | 0.90 | | $35.16 \pm 2.96$ | $34.94 \pm 1.01$ |

## D.5 ADDITIONAL PATTERN ANALYSIS

In FrozenLake and TwoRooms-Trap, trap states can prematurely terminate episodes, leading to optimal sets focusing on avoiding trap states as well as reaching the goal. In CliffWalk, the large penalty for falling into the cliff causes optimal sets to include off-path states adjacent to the cliff, effectively constraining the agent's behavior. These effects are accentuated by the reward imputation strategy in UDS (Yu et al., 2022), which assumes unlabeled states have zero reward. Further ablation with alternative settings (e.g., Q-truncated) is shown in Appendix D.9.

To better understand the effectiveness of heuristic strategies in Breakout, we further analyze the state sets selected by `visitation` and `uniform` methods. As shown in Figure 3, `visitation` consistently outperforms `uniform` across all budget levels. To investigate this, we sampled 100 state sets from each strategy and calculated the cumulative reward present within the selected states.

Table 9 shows the average sum of rewards across these samples at varying feedback levels. The results indicate that state sets selected by `visitation` heuristics consistently contain a higher concentration of high-reward states compared to `uniform`. In Breakout, high-reward states often correspond to frames where the paddle is well-aligned with the ball to prevent it from being lost, which yields a reward of 1. The `visitation` heuristic is biased toward such frequently encountered high-value configurations during data collection, whereas `uniform` sampling provides more dispersed but less reward-focused coverage.

This quantitative observation directly supports the qualitative interpretation of the performance gap seen in Figure 3: `visitation`'s tendency to prioritize paddle-ball alignment states leads to a higher sum of rewards in the labeled dataset and therefore facilitates better value propagation during offline RL training.

Table 9: Sum of rewards in the state sets selected by `visitation` and `uniform` heuristics on Breakout. At each feedback level, we sample 100 state sets and report the mean ($\pm$ standard error) of total rewards present in the selected states. Higher values for `visitation` indicate its stronger tendency to select high-reward (paddle-ball alignment) states.

| Percentage Feedback | visitation | uniform |
|---|---|---|
| 0.146 | $60936.980 \pm 49.963$ | $10039.340 \pm 368.025$ |
| 0.291 | $65967.740 \pm 15.982$ | $20394.690 \pm 514.998$ |
| 0.437 | $67718.620 \pm 9.163$ | $30736.510 \pm 609.436$ |
| 0.583 | $68640.250 \pm 4.266$ | $39807.620 \pm 668.318$ |
| 0.728 | $69182.230 \pm 2.760$ | $51333.890 \pm 522.632$ |
| 0.874 | $69522.340 \pm 1.531$ | $61671.510 \pm 347.499$ |

## D.6 BEHAVIOUR POLICY DEPENDENCE

In the experiments below, we constructed five offline datasets in the Breakout domain using five different mixture ratios of an expert policy and a uniform-random policy, where the mixture ratio indicates the probability of following the expert policy (e.g., 0.3 means 30% expert and 70% random). We evaluate the `guided` selection strategy with both UDS (Section 2) and Adaptive Q-learning (Appendix D.9), confirming the Section 4.1 conclusion on coverage-driven performance across different datasets and including a Behavior Cloning (BC) comparison under Adaptive Q-learning.

The table below summarizes the number of unique states visited under each behavior policy ratio, which directly reflects dataset coverage:

Table 10: Unique states under different expert policy ratios.

| Expert Policy Ratio | 0.0 | 0.3 | 0.5 | 0.7 | 1.0 |
|---|---|---|---|---|---|
| Unique States | 99,254 | 26,255 | 13,731 | 7,110 | 3,535 |

For `guided` with UDS, we compare selective reward-labeling with a uniform-random baseline at 0%. UDS imputes all unlabeled rewards as zero, making uniform coverage an appropriate comparison for its behavior in the absence of reward labels. Guided selection quickly outperforms the naive

uniform-random baseline even at small labeling budgets, confirming the conclusion from Section 4.1, showing a robust trend across diverse datasets.

Table 11: Performance of different feedback levels under varying behavior policies with UDS.

| Behavior Policy | Uniform-random (0%) | 15% | 30% | 45% | 60% | 75% | 90% | IQL (100%) |
|---|---|---|---|---|---|---|---|---|
| 0.0 | $0.77 \pm 0.06$ | $0.51 \pm 0.03$ | $0.57 \pm 0.03$ | $0.53 \pm 0.04$ | $0.56 \pm 0.04$ | $0.70 \pm 0.05$ | $0.87 \pm 0.05$ | $0.84 \pm 0.06$ |
| 0.3 | $0.77 \pm 0.06$ | $16.27 \pm 0.34$ | $16.31 \pm 0.37$ | $17.53 \pm 0.38$ | $17.98 \pm 0.34$ | $17.83 \pm 0.37$ | $17.70 \pm 0.38$ | $17.82 \pm 0.35$ |
| 0.5 | $0.77 \pm 0.06$ | $7.13 \pm 0.11$ | $14.12 \pm 0.34$ | $17.39 \pm 0.29$ | $17.60 \pm 0.33$ | $17.17 \pm 0.35$ | $17.06 \pm 0.37$ | $17.75 \pm 0.22$ |
| 0.7 | $0.77 \pm 0.06$ | $4.26 \pm 0.03$ | $5.74 \pm 0.07$ | $6.44 \pm 0.11$ | $7.14 \pm 0.10$ | $7.27 \pm 0.10$ | $7.60 \pm 0.10$ | $7.59 \pm 0.12$ |
| 1.0 | $0.77 \pm 0.06$ | $3.24 \pm 0.03$ | $3.64 \pm 0.04$ | $5.51 \pm 0.05$ | $5.59 \pm 0.06$ | $5.64 \pm 0.06$ | $5.49 \pm 0.06$ | $5.60 \pm 0.08$ |

For `guided` with Adaptive Q-learning, we include Behavior Cloning (BC) at 0% as a reference point, representing fully supervised learning from raw trajectories without reward labeling. Guided selection with even small labeling budgets consistently outperforms BC—except when using datasets collected purely by uniform-random policies. These results also support the Section 4.1 conclusion.

Table 12: Performance of different feedback levels under varying behavior policies with Adaptive Q-learning.

| Behavior Policy | BC (0%) | 15% | 30% | 45% | 60% | 75% | 90% | IQL (100%) |
|---|---|---|---|---|---|---|---|---|
| 0.0 | $0.77 \pm 0.06$ | $0.48 \pm 0.03$ | $0.61 \pm 0.05$ | $0.81 \pm 0.04$ | $0.87 \pm 0.05$ | $0.87 \pm 0.03$ | $0.85 \pm 0.06$ | $0.82 \pm 0.05$ |
| 0.3 | $2.77 \pm 1.04$ | $16.53 \pm 0.36$ | $16.37 \pm 0.38$ | $16.88 \pm 0.36$ | $16.89 \pm 0.36$ | $16.97 \pm 0.35$ | $17.17 \pm 0.36$ | $17.77 \pm 1.04$ |
| 0.5 | $5.23 \pm 0.46$ | $14.12 \pm 0.34$ | $17.39 \pm 0.29$ | $17.60 \pm 0.33$ | $16.17 \pm 0.35$ | $17.06 \pm 0.37$ | $17.75 \pm 0.22$ | $17.49 \pm 0.66$ |
| 0.7 | $4.89 \pm 0.07$ | $5.71 \pm 0.07$ | $6.14 \pm 0.09$ | $6.69 \pm 0.10$ | $7.23 \pm 0.11$ | $7.10 \pm 0.10$ | $7.48 \pm 0.11$ | $7.98 \pm 0.38$ |
| 1.0 | $3.70 \pm 0.05$ | $4.31 \pm 0.05$ | $5.11 \pm 0.05$ | $5.47 \pm 0.06$ | $5.34 \pm 0.06$ | $5.52 \pm 0.06$ | $5.54 \pm 0.06$ | $5.46 \pm 0.21$ |

---

**Takeaway**

1. The results above demonstrate that the learned policy does not degenerate to the behavior policy simply because frequently visited states are labeled. Instead, state coverage of the offline dataset is a dominant factor in final policy performance. Even when the behavior policy is weak, guided reward selection enables strong policy learning as long as coverage is sufficient.

2. Behavior Cloning (BC) is generally outperformed by guided reward selection with only 15% feedback. This confirms that the outcome of reward selection does not simply imitate the behavior policy, and that guided selection can leverage a small number of labeled rewards to learn a policy that outperforms BC, even when dataset coverage is limited, and performs even better when coverage is sufficient.

---

### D.7 INITIAL SAMPLE SENSITIVITY

To evaluate how the method responds to different initial state subset selections, we performed initial sample ratio experiments: a fraction of the total labeling budget was randomly allocated at the start, after which the remaining budget was spent according to the `guided` selection strategy (1st row). For comparison, we also evaluated two static baselines, `ES 50` (2nd row) and `ES 200` (3rd row), under the same initial sample ratios. Results are reported for both a dense reward domain (`Graph`, Table 13) and a sparse reward domain (`TwoRooms`, Table 14).

Across both domains, we observe that the optimal selection methods (`ES 200` and `ES 50`) generally outperform `guided` selection at equivalent budgets, as they are not penalized by suboptimal early queries and have access to the evaluator. However, the effect of initial random sampling differs by domain. In the sparse reward TwoRooms environment, larger initial samples improve early performance because random initialization has a better chance of labeling terminal states, which accelerates learning once `guided` selection begins. In contrast, in the dense reward Graph environment, larger initial samples degrade performance, as they waste labeling budget on states that `guided` selection would have efficiently deprioritized. These results suggest that guided selection still converges to near-optimal performance once it has enough budget, even after suboptimal initial state selections.

Table 13: Results across different initial sample ratios and feedback percentages on Graph domain with `guided` selection strategy.

| Initial Sample Ratio | 0.1 | 0.3 | 0.5 | 0.7 | 0.9 |
|---|---|---|---|---|---|
| 0 | $3.701 \pm 0.129$ | $5.831 \pm 0.137$ | $7.110 \pm 0.099$ | $7.830 \pm 0.040$ | $8.000 \pm 0.000$ |
| | $5.001 \pm 0.000$ | $6.000 \pm 0.000$ | $7.003 \pm 0.000$ | $8.000 \pm 0.000$ | $8.000 \pm 0.000$ |
| | $7.003 \pm 0.000$ | $8.000 \pm 0.000$ | $8.000 \pm 0.000$ | $8.000 \pm 0.000$ | $8.000 \pm 0.000$ |
| 0.1 | $3.406 \pm 0.135$ | $5.699 \pm 0.130$ | $7.041 \pm 0.094$ | $7.731 \pm 0.049$ | $8.000 \pm 0.000$ |
| | $3.750 \pm 0.140$ | $5.311 \pm 0.128$ | $7.101 \pm 0.121$ | $7.532 \pm 0.116$ | $8.000 \pm 0.000$ |
| | $4.120 \pm 0.146$ | $6.391 \pm 0.127$ | $7.081 \pm 0.109$ | $7.501 \pm 0.114$ | $8.000 \pm 0.000$ |
| 0.3 | – | $4.932 \pm 0.164$ | $6.531 \pm 0.117$ | $7.581 \pm 0.062$ | $8.000 \pm 0.000$ |
| | – | $6.011 \pm 0.092$ | $6.852 \pm 0.079$ | $7.401 \pm 0.066$ | $8.000 \pm 0.000$ |
| | – | $6.111 \pm 0.087$ | $7.172 \pm 0.063$ | $7.541 \pm 0.057$ | $8.000 \pm 0.000$ |
| 0.5 | – | – | $6.234 \pm 0.114$ | $7.399 \pm 0.066$ | $8.000 \pm 0.000$ |
| | – | – | $6.502 \pm 0.084$ | $7.002 \pm 0.065$ | $8.000 \pm 0.000$ |
| | – | – | $7.102 \pm 0.067$ | $7.451 \pm 0.059$ | $8.000 \pm 0.000$ |

Table 14: Results across different initial sample ratios and feedback percentages on the TwoRooms domain with `guided` selection strategy.

| Initial Sample Ratio | 0.1 | 0.3 | 0.5 | 0.7 | 0.9 |
|---|---|---|---|---|---|
| 0 | $0.012 \pm 0.010$ | $0.077 \pm 0.027$ | $0.173 \pm 0.039$ | $0.270 \pm 0.046$ | $0.732 \pm 0.046$ |
| | $1.000 \pm 0.000$ | $1.000 \pm 0.000$ | $1.000 \pm 0.000$ | $1.000 \pm 0.000$ | $1.000 \pm 0.000$ |
| | $1.000 \pm 0.000$ | $1.000 \pm 0.000$ | $1.000 \pm 0.000$ | $1.000 \pm 0.000$ | $1.000 \pm 0.000$ |
| 0.1 | $0.221 \pm 0.041$ | $0.251 \pm 0.043$ | $0.331 \pm 0.047$ | $0.511 \pm 0.050$ | $0.830 \pm 0.038$ |
| | $0.321 \pm 0.047$ | $0.571 \pm 0.049$ | $0.700 \pm 0.046$ | $0.870 \pm 0.034$ | $1.000 \pm 0.000$ |
| | $0.451 \pm 0.050$ | $0.770 \pm 0.042$ | $0.950 \pm 0.022$ | $0.970 \pm 0.017$ | $1.000 \pm 0.000$ |
| 0.3 | – | – | $0.601 \pm 0.049$ | $0.700 \pm 0.046$ | $0.880 \pm 0.032$ |
| | – | – | $0.910 \pm 0.029$ | $0.980 \pm 0.014$ | $1.000 \pm 0.000$ |
| | – | – | $0.950 \pm 0.022$ | $0.990 \pm 0.010$ | $1.000 \pm 0.000$ |
| 0.5 | – | – | – | $0.780 \pm 0.041$ | $0.920 \pm 0.027$ |
| | – | – | – | $0.850 \pm 0.036$ | $1.000 \pm 0.000$ |
| | – | – | – | $0.890 \pm 0.031$ | $1.000 \pm 0.000$ |

**Takeaway**

The impact of initial state subset selection depends on the reward structure. In sparse reward settings, allocating more initial random labels can improve early performance by increasing the chance of covering terminal states, whereas in dense reward settings, it can hinder performance by diverting labels away from more informative regions. While `guided` selection may lag behind optimal methods a lot in the early stage, it recovers as more budget becomes available and converges to strong final policies.

## D.8 TRADEOFF SCHEDULES

The `guided` strategy gradually shifts from **exploration** to **exploitation**. This shift is controlled by a **decay function** and related parameters:

1. **Decay function** determines how quickly exploration weight decreases over the course of the labeling budget.

   - **Linear decay:** exploration weight decreases at a constant rate from start to finish.
   - **Convex decay:** exploration weight decreases quickly at the start and then flattens out, prioritizing exploitation early.

- **Concave decay:** exploration weight decreases slowly at the start and then drops quickly near the end, emphasizing exploration for longer before rapidly switching to exploitation.

2. **Decay temperature** controls how sharp or gentle the decay curve is for convex and concave schedules. A larger temperature means a steeper initial drop (for convex) or a flatter early phase (for concave).

3. **Fixed time threshold** specifies the fraction of total iterations after which exploration stops entirely, forcing the strategy to fully exploit. For example, `fixtime = 0.7` means exploration will stop only after rewards have been queried for at least 70% of all rewards in the dataset; if the total budget is smaller than that threshold, exploration is never fully turned off.

4. **Initial sample size** determines, respectively, how many states are chosen randomly before guided selection begins and how unqueried rewards are treated.

> **Takeaway**
>
> These parameters define how the `guided` strategy balances exploration and exploitation over time. We performed a combinational search over these settings and selected a configuration that provided good performance, which we use as the default in our experiments.

### D.9   A DIFFERENT POLICY UPDATE RULE

As illustrated in Figure 2, the core of this work is to propose and compare different reward selection strategies, which should be applicable to any `Alg`. While our main results focus on using UDS, in this section we apply the same selection strategies to an alternative `Alg` we propose.

#### D.9.1   ADAPTED Q-LEARNING

We use Q-learning—a value-based algorithm variants of which are widely used in offline settings (Levine et al., 2020; Kostrikov et al., 2021)—for policy updates in `Alg`. However, missing reward labels for some samples in `RLLF` pose a challenge: *how should the policy be updated when samples without rewards are encountered?* While assumptions might be made to facilitate modeling of unknown rewards, those reward estimates may be arbitrarily incorrect, especially in discrete domains.

Consequently, for states where rewards are unavailable (i.e., $s \notin \mathcal{S}_{[B]}$), we make no assumptions and treat the reward as being *undefined*. As a result, this algorithm sets unknown Q-values to zero, in contrast the UDS algorithm sets unknown reward values to zero. This approach aligns with the principle of *pessimism* in offline RL, which ensures that potentially erronous value estimates from *unseen* data are not used to update values of *seen* data—a strategy whose benefits are widely studied (Jin et al., 2021; Xie et al., 2021). To accommodate undefined rewards, we modify the vanilla Q-learning update rule as follows:

$$
Q(s,a) \longleftarrow
\begin{cases}
Q(s,a) + \alpha \left( r(s,a) + \gamma * \max_{a'} Q(s',a') - Q(s,a) \right), & s \in \mathcal{S}_{[B]} \;\&\; s' \in \mathcal{S}_{[B]} \\
\alpha\, r(s,a), & s \in \mathcal{S}_{[B]} \;\&\; s' \notin \mathcal{S}_{[B]} \\
\underbrace{\text{undefined}}_{=0}, & s \notin \mathcal{S}_{[B]}
\end{cases}
\tag{6}
$$

For $B = |\mathcal{S}|$, i.e., when all rewards are known for all states, this reduces to the standard Q-learning update rule (Sutton & Barto, 2018). For $B < |\mathcal{S}|$, this update rule yields a *truncated* estimate of the standard Q-values, with a corresponding *truncated* Bellman operator. To distinguish these Q-values from the standard definition, we use $\tilde{Q}$ to denote Q-values estimated from the update rule in Equation 6.

The values $\tilde{Q}(s,a)$ are only defined for states $s \in \mathcal{S}_{[B]}$. Consequently, a greedy policy derived from the truncated Q-values can only be defined for $s \in \mathcal{S}_{[B]}$. For states $s \notin \mathcal{S}_{[B]}$, there is no reward feedback is available and $\tilde{Q}(s,a)$ is undefined, and we cannot evaluate the varying effects of actions in those states. In the absence of any evaluative signal for actions, we default to the data collecting policy $\pi_D$ at those states.

$$
\pi_{[B]} = \pi_{[\mathcal{S}_{[B]}]} =
\begin{cases}
\arg\max_a \tilde{Q}(s,a), & s \in \mathcal{S}_{[B]} \\
\pi_D, & s \notin \mathcal{S}_{[B]}
\end{cases}
\tag{7}
$$

This update scheme is denoted by `Alg`, and the policy output by `Alg`$(\mathcal{D}, \mathcal{S}_{[B]})$ is denoted by $\pi_{[B]}$, or equivalently, $\pi_{[\mathcal{S}_{[B]}]}$ when emphasizing the dependence on $\mathcal{S}_{[B]}$. Policy updates only occur at states $s \in \mathcal{S}_{[B]}$. Selecting a set of states to label with reward amounts determines states at which the policy gets updated—potentially to differ from the data-collecting policy—and the strategy for selecting these states $\mathcal{Q}^{(B)}$ to optimize Equation (1) is the focus of the following sections.

#### D.9.2   PERFORMANCE OF HEURISTICS SELECTION STRATEGY

We evaluate `guided`, `visitation`, and `uniform` selection strategies under Adaptive Q-Learning on small domains as shown in the Table 15. The trends largely align with the findings in the main text and remain consistent with those observed under UDS. In domains such as Graph, Tree, CliffWalk, and TwoRooms-Trap, where the optimal policy follows a narrow set of trajectories, path-following methods (`guided` and `visitation`) perform best. In contrast, TwoRooms and FrozenLake contain multiple viable paths to the goal, making broader state coverage more advantageous; here, `uniform` selection achieves superior results. Adaptive Q-Learning confirms the strong dependence of heuristic effectiveness on domain characteristics, including transition determinism, reward sparsity, and bottleneck structures (as discussed in Section 4.1).

Table 15: Comparison of `guided`, `visitation`, and `uniform` heuristic selection strategies on prototypical domains. For each domain, the table presents the mean policy return ($\pm$ standard error) and the corresponding optimality gap (in parentheses) across five percentage feedback levels.

| Domains | Percentage Feedback | guided | visitation | uniform |
|---|---|---|---|---|
| Graph | 0.1 | **4.477 ± 0.040 (0.860)** | 4.397 ± 0.036 (0.940) | 4.171 ± 0.040 (1.166) |
| | 0.3 | **5.616 ± 0.069 (1.549)** | 5.480 ± 0.068 (1.685) | 5.048 ± 0.062 (2.117) |
| | 0.5 | **6.604 ± 0.098 (1.396)** | 6.385 ± 0.101 (1.615) | 5.697 ± 0.081 (2.303) |
| | 0.7 | **7.502 ± 0.086 (0.498)** | 7.229 ± 0.093 (0.771) | 6.019 ± 0.127 (1.981) |
| | 0.9 | **8.000 ± 0.000 (0.000)** | **8.000 ± 0.000 (0.000)** | **8.000 ± 0.000 (0.000)** |
| Tree | 0.1 | **8.300 ± 0.144 (3.424)** | 8.059 ± 0.116 (3.665) | 6.753 ± 0.076 (4.971) |
| | 0.3 | **13.317 ± 0.337 (3.608)** | 12.126 ± 0.238 (4.798) | 8.484 ± 0.134 (8.440) |
| | 0.5 | **16.120 ± 0.183 (1.340)** | 14.917 ± 0.277 (2.543) | 10.445 ± 0.240 (7.014) |
| | 0.7 | **17.354 ± 0.041 (0.269)** | 16.870 ± 0.151 (0.753) | 11.637 ± 0.343 (5.985) |
| | 0.9 | **17.689 ± 0.012 (0.030)** | 17.675 ± 0.013 (0.016) | 16.280 ± 0.292 (1.379) |
| CliffWalk | 0.1 | −414.059 ± 7.923 (171.814) | −414.059 ± 7.923 (171.814) | −488.198 ± 6.642 (245.953) |
| | 0.3 | **−236.441 ± 18.131 (136.441)** | −237.081 ± 18.171 (137.081) | −433.176 ± 15.181 (333.176) |
| | 0.5 | −155.088 ± 13.893 (55.088) | **−154.042 ± 13.888 (54.042)** | −409.481 ± 20.146 (309.481) |
| | 0.7 | −123.490 ± 7.651 (92.459) | **−100.437 ± 0.881 (69.406)** | −378.334 ± 24.350 (347.302) |
| | 0.9 | −146.676 ± 11.375 (132.023) | **−107.590 ± 5.313 (92.937)** | −341.785 ± 27.414 (327.131) |
| FrozenLake | 0.1 | 0.024 ± 0.000 (0.010) | **0.024 ± 0.000 (0.010)** | 0.024 ± 0.000 (0.010) |
| | 0.3 | 0.024 ± 0.000 (0.048) | 0.024 ± 0.000 (0.048) | **0.025 ± 0.001 (0.047)** |
| | 0.5 | 0.024 ± 0.000 (0.222) | 0.023 ± 0.000 (0.223) | **0.027 ± 0.001 (0.218)** |
| | 0.7 | 0.073 ± 0.015 (0.595) | 0.036 ± 0.007 (0.631) | **0.098 ± 0.014 (0.569)** |
| | 0.9 | **0.374 ± 0.030 (0.336)** | 0.267 ± 0.025 (0.443) | 0.368 ± 0.026 (0.341) |
| TwoRooms | 0.1 | 0.025 ± 0.001 (0.289) | 0.025 ± 0.001 (0.289) | **0.030 ± 0.001 (0.283)** |
| | 0.3 | 0.013 ± 0.001 (0.939) | 0.012 ± 0.001 (0.939) | **0.033 ± 0.003 (0.919)** |
| | 0.5 | 0.007 ± 0.000 (0.992) | 0.008 ± 0.000 (0.992) | **0.043 ± 0.005 (0.956)** |
| | 0.7 | 0.159 ± 0.035 (0.841) | 0.085 ± 0.027 (0.915) | **0.230 ± 0.034 (0.770)** |
| | 0.9 | 0.721 ± 0.044 (0.279) | **0.761 ± 0.043 (0.239)** | 0.720 ± 0.042 (0.280) |
| TwoRooms-Trap | 0.1 | **−55.947 ± 0.920 (32.444)** | −57.720 ± 0.628 (34.217) | −62.899 ± 0.487 (39.396) |
| | 0.3 | **−41.188 ± 1.123 (39.950)** | −44.392 ± 0.832 (43.154) | −53.528 ± 0.692 (52.290) |
| | 0.5 | **−14.334 ± 0.837 (14.872)** | −21.868 ± 0.894 (22.406) | −40.030 ± 0.837 (40.568) |
| | 0.7 | **−0.178 ± 0.057 (1.176)** | −1.001 ± 0.332 (1.999) | −26.138 ± 0.966 (27.136) |
| | 0.9 | **1.000 ± 0.000 (0.000)** | **1.000 ± 0.000 (0.000)** | −4.577 ± 0.689 (5.577) |

### D.9.3 Performance of Optimal Selection Strategy

We evaluate `brute-force`, `sequential-greedy`, `ES 200`, and `ES 50` under Adaptive Q-Learning with the same setting as in the main text shown in Table 16. The findings closely mirror those observed with UDS. `Sequential-greedy` consistently matches the performance of `brute-force`, validating its effectiveness as a scalable approximation to the true optimal state set. `ES 200` reliably outperforms `ES 50`, and both evolutionary variants generally exceed the performance of `guided` selection at moderate to high budgets. These results reaffirm the relative ordering and conclusions reported in the main text, demonstrating that the effectiveness of optimized selection strategies remains stable across different policy learning algorithms.

Table 16: Performance comparison of `brute-force`, `sequential-greedy`, and `ES` on prototypical domains. Results are reported on training datasets, with test performance shown in parentheses (e.g., train score (test score)). Test scores are reported as mean $\pm$ standard error across five test datasets. `ES 200` corresponds to $k = 10, m = 20$ and `ES 50` to $k = 10, m = 5$.

| Domains | Percentage Feedback | brute-force | sequential-greedy | ES 200 | ES 50 | guided |
|---|---|---|---|---|---|---|
| Graph | 0.1 | 5.337(3.032 ± 0.213) | 5.337(3.032 ± 0.213) | 5.308(3.014 ± 0.211) | 4.214(1.521 ± 0.226) | 4.477 |
| | 0.3 | 7.165(6.004 ± 0.128) | 7.165(6.004 ± 0.128) | 7.157(5.994 ± 0.124) | 6.275(4.518 ± 0.164) | 5.616 |
| | 0.5 | 8.000(8.000 ± 0.000) | 8.000(8.000 ± 0.000) | 8.000(8.000 ± 0.000) | 6.589(5.256 ± 0.115) | 6.604 |
| | 0.7 | 8.000(8.000 ± 0.000) | 8.000(8.000 ± 0.000) | 8.000(8.000 ± 0.000) | 8.000(8.000 ± 0.000) | 7.502 |
| | 0.9 | 8.000(8.000 ± 0.000) | 8.000(8.000 ± 0.000) | 8.000(8.000 ± 0.000) | 8.000(8.000 ± 0.000) | 8.000 |
| Tree | 0.1 | 11.724(8.092 ± 0.276) | 11.724(8.092 ± 0.276) | 11.724(8.092 ± 0.276) | 9.073(4.078 ± 0.384) | 8.300 |
| | 0.3 | 16.925(16.349 ± 0.056) | 16.925(16.349 ± 0.056) | 13.282(10.283 ± 0.238) | 9.637(5.187 ± 0.334) | 13.317 |
| | 0.5 | 17.460(17.406 ± 0.017) | 17.460(17.406 ± 0.017) | 17.235(16.982 ± 0.025) | 12.656(9.909 ± 0.184) | 16.120 |
| | 0.7 | 17.623(17.627 ± 0.006) | 17.623(17.627 ± 0.006) | 17.513(17.489 ± 0.009) | 15.217(13.324 ± 0.142) | 17.354 |
| | 0.9 | 17.659(17.788 ± 0.001) | 17.659(17.788 ± 0.001) | 17.678(17.777 ± 0.000) | 17.655(17.728 ± 0.001) | 17.689 |
| CliffWalk | 0.1 | −242.245(−231.272 ± 6.042) | −242.245(−231.272 ± 6.042) | −322.823(−347.828 ± 12.005) | −409.384(−414.365 ± 26.537) | −414.059 |
| | 0.3 | −100.000(−100.000 ± 0.000) | −100.000(−100.000 ± 0.000) | −150.081(−150.586 ± 4.002) | −320.748(−308.868 ± 13.555) | −236.441 |
| | 0.5 | −100.000(−100.000 ± 0.000) | −100.000(−100.000 ± 0.000) | −100.000(−100.000 ± 0.000) | −180.969(−189.833 ± 3.670) | −155.088 |
| | 0.7 | −31.031(−31.142 ± 1.045) | −31.031(−31.142 ± 1.045) | −100.000(−100.000 ± 0.000) | −186.756(−180.828 ± 8.232) | −123.490 |
| | 0.9 | −14.653(−14.506 ± 0.138) | −14.653(−14.506 ± 0.138) | −100.000(−100.000 ± 0.000) | −100.000(−100.000 ± 0.000) | −146.676 |
| FrozenLake | 0.1 | 0.034(0.032 ± 0.001) | 0.034(0.032 ± 0.001) | 0.031(0.030 ± 0.000) | 0.031(0.030 ± 0.001) | 0.024 |
| | 0.3 | 0.072(0.049 ± 0.003) | 0.072(0.049 ± 0.003) | 0.036(0.032 ± 0.001) | 0.032(0.034 ± 0.001) | 0.024 |
| | 0.5 | 0.246(0.347 ± 0.029) | 0.246(0.347 ± 0.029) | 0.067(0.057 ± 0.004) | 0.054(0.045 ± 0.005) | 0.024 |
| | 0.7 | 0.667(0.629 ± 0.011) | 0.667(0.629 ± 0.011) | 0.199(0.212 ± 0.006) | 0.196(0.223 ± 0.008) | 0.073 |
| | 0.9 | 0.710(0.688 ± 0.013) | 0.710(0.688 ± 0.013) | 0.679(0.703 ± 0.006) | 0.699(0.709 ± 0.013) | 0.374 |
| TwoRooms | 0.1 | 0.314(0.321 ± 0.031) | 0.314(0.321 ± 0.031) | 0.063(0.073 ± 0.014) | 0.038(0.046 ± 0.009) | 0.025 |
| | 0.3 | 0.952(0.952 ± 0.005) | 0.952(0.952 ± 0.005) | 0.310(0.314 ± 0.029) | 0.052(0.057 ± 0.009) | 0.013 |
| | 0.5 | 1.000(1.000 ± 0.000) | 1.000(1.000 ± 0.000) | 0.365(0.362 ± 0.026) | 0.270(0.270 ± 0.022) | 0.007 |
| | 0.7 | 1.000(1.000 ± 0.000) | 1.000(1.000 ± 0.000) | 1.000(1.000 ± 0.000) | 0.999(1.000 ± 0.000) | 0.159 |
| | 0.9 | 1.000(1.000 ± 0.000) | 1.000(1.000 ± 0.000) | 1.000(1.000 ± 0.000) | 1.000(1.000 ± 0.000) | 0.721 |
| TwoRooms-Trap | 0.1 | −23.503(−23.047 ± 0.319) | −23.503(−23.047 ± 0.319) | −31.646(−32.431 ± 0.736) | −53.449(−53.509 ± 0.204) | −55.947 |
| | 0.3 | −1.238(−1.243 ± 0.017) | −1.238(−1.243 ± 0.017) | −11.259(−10.935 ± 0.174) | −35.621(−35.996 ± 0.556) | −41.188 |
| | 0.5 | 0.538(0.540 ± 0.016) | 0.538(0.540 ± 0.016) | −0.845(−0.793 ± 0.030) | −17.590(−17.505 ± 0.139) | −14.334 |
| | 0.7 | 0.998(0.998 ± 0.000) | 0.998(0.998 ± 0.000) | −0.233(−0.258 ± 0.024) | −14.739(−14.826 ± 0.245) | −0.178 |
| | 0.9 | 1.000(1.000 ± 0.000) | 1.000(1.000 ± 0.000) | 1.000(1.000 ± 0.000) | 0.862(0.845 ± 0.010) | 1.000 |

