# OpenReview forum: "Which Rewards Matter? Reward Selection for Reinforcement Learning under Limited Feedback"
_ICLR.cc/2026/Conference — Submitted to ICLR 2026_

### Official Review · Reviewer_pvN8 · 2025-10-26

**Soundness:** 4
**Presentation:** 2
**Contribution:** 3
**Rating:** 8
**Confidence:** 3

**Summary:**

The authors tackle the problem of reward selection in reinforcement learning from limited feedback within the context of Markov Decision Processes with discrete state and action spaces. They focus on an offline setting, where a static dataset of tuples (state, action, next state) is available, but no reward labels. Under the assumption that obtaining labels (e.g., from human feedback) involves a significant cost, the question arises which samples should receive a reward label to obtain the best possible policy. The authors translate this to the problem of selecting the subset of states of a given cardinality that yields the best policy. After formalizing this task, they suggest both heuristics and learning-based solution approaches, which are either iterative or choose all relevant states at the same time. The approaches are compared on several prototypical and four MinAtar domains. Finally, the authors present some insights on patterns that evolve in all successful state selection strategies.

**Strengths:**

- The problem presented is interesting, and the formalization of the optimal state selection independent of questions of state reachability and exploration seems unexplored, while relevant.
- The heuristics and training-based approaches present interesting starting points for solving the presented problem. Evaluation of the approaches is overall well done and statistically significant.
- The observed patterns in state selection provide valuable insights for devising task-specific heuristics or new selection algorithms.

**Weaknesses:**

- The proposed problem formalization and solution approaches are mainly devised for discrete state spaces. Many of the presented motivational use cases involve continuous state spaces. Either the motivation or the methodology should be improved. Also, problem complexity will increase further for continuous action spaces. It would be valuable to discuss this at some point in the paper.
- Presentation could be improved by making figures 4 and 5 larger. Figure 4 is not really discussed in the text and it does not become clear why it cannot simply be integrated with Table 1. In Figure 3, there seems to be come mistake, as a wall clock time of 1e9 days or higher seems rather unlikely.

**Questions:**

- In Eq. (4), do you mean $\hat{d}^{\pi_d}_\text{prev}$?
- In Fig. (4), why are the on-policy variants of the 'visitation' and 'guided' heuristic not presented?
- Are any of the proposed solution approaches applicable to continuous state spaces?

---

> ### Author Response · Authors · 2025-11-20
>
> Thank you for your review! We appreciate your positive assessment of our work, and in particular your summary, which captures the core ideas and contributions of the paper accurately.
>
>
> We respond to weaknesses below.
>
> > Motivation and methodology for continuous state spaces
>
> It would certainly be interesting to extend these methods to continuous state spaces. Our motivation to focus on discrete state spaces is two-part: (1) several of the motivating examples, such as LLM post-training and AI-driven drug discovery, involve *discrete* state and action spaces making our analysis directly relevant to those use cases, and (2) working with discrete states allows us to obtain insights and takeaways that are *more interpretable*, particularly regarding which states are high-impact and constitute the optimal labeled subset.
>
> Methodology for continuous states: Selection in those cases would strongly be influenced by the **state-action representations**, since there exists a degree of generalization across similar states and actions in both the reward and transition functions unlike that in the discrete case. This can sometimes make the problem easier. For example one of the papers we reference, Zhan et al. [1], leverages exactly this artifact by considering MDPs with linear reward representations.
>
> > Improving presentation of Figures 3, 4, 5
>
> Thank you for the suggestions regarding figure presentation -- we have enlarged the figures and improve their references in the main text.
> Regarding the concern about “1e9 days’’  in Figure 3: this number is an estimated wall-clock time, not an actually measured runtime. The estimate follows from the theoretical training-cost of brute-force search. As noted in the paper, even for a modest domain with $|S|=50$ and budget $B=25$, brute-force evaluation requires $\binom{50}{25} \approx 10^{14}$ calls to the evaluator. On an RTX 2080 Ti, we measured a throughput of roughly 2,000 evaluator calls per minute, which implies that a full brute-force sweep would take on the order of $10^{14} / (365 \times 24 \times 60 \times 2000) \approx 10^5$ years. For the Seaquest domains at 60% feedback, the numbers are significantly higher. Those extremely large values on the time axis are a reflect of the true combinatorial cost of brute-force search. In the revision, we have relabelled the axis as **estimated wall-clock time (log scale)** and clarify on [Line 384] that the values are extrapolated estimates rather than empirical measurements.
>
> # Questions
>
> Next, we answer the questions one-by-one.
>
> > In Eq. (4), do you mean $\hat{d}^{\pi_d}_\text{prev}$?
>
> Yes indeed, thanks for catching that typo. It has been updated.
>
> > In Fig. (4), why are the on-policy variants of the 'visitation' and 'guided' heuristic not presented?
>
> In smaller tabular domains, to compute the on-policy state visitation distribution, we estimate the transition dynamics for on-policy methods by counting empirical transitions in the offline dataset. This produces a workable approximation of $\widehat{p}(s, a, s')$, allowing computation of approximate on-policy visitation distributions for results in Table 1.
> In contrast, for high-dimensional domains such as in MinAtar (Figure 4), this approach is no longer viable. Counting alone does not yield an accurate transition model because the state space is extremely large, many state–action pairs are visited only once (or not at all), and empirical counts are too sparse to approximate dynamics meaningfully.
>
> >   Are any of the proposed solution approaches applicable to continuous state spaces?
>
> Discussed under "Motivation and methodology for continuous state spaces" above.
>
> We hope these clarifications address your concerns. Please let us know if any further details would be helpful.
>
> ---
> [1] Zhan, Wenhao, et al. "How to Query Human Feedback Efficiently in RL?." (2023).

---

### Official Review · Reviewer_XCoh · 2025-10-27

**Soundness:** 2
**Presentation:** 2
**Contribution:** 3
**Rating:** 4
**Confidence:** 3

**Summary:**

This paper introduces and formalizes a new problem setting called Reward Selection for Reinforcement Learning under Limited Feedback (RLLF). The core idea is that rewards are expensive to obtain in many real-world applications such as RLHF for large language models or molecule discovery. Therefore, instead of assuming all states have labeled rewards, the authors consider a constrained setting where only a limited number of states can be reward-labeled.
The key research question is: Which subset of states should be labeled with rewards to maximize the performance of the resulting policy?
The paper proposes both heuristic (training-free) and training-based reward selection strategies. Experiments include simple synthetic scenarios and larger benchmarks (MinAtar).

**Strengths:**

1. The concept of "reward selection" is good, and fills a gap between active RL (which couples exploration and reward querying) and active reward modeling (which focuses on reward estimation rather than policy performance).

2. The authors argue the importance of limited-feedback RL in settings like RLHF, where feedback cost dominates computational cost. Furthermore, the experiments are well-organized and multi-scale. The inclusion of both synthetic toy domains and more realistic environments (MinAtar) helps generalize findings.

**Weaknesses:**

1. The work is entirely empirical. There is no formal analysis of why certain reward selections yield near-optimal policies, or whether reward selection admits approximation guarantees (e.g., submodularity, monotonicity). Without such theory, claims of near-optimality remain heuristic.

2. The training-phase setup assumes access to an "oracle evaluator", that can compute policy returns under the true reward function. This is a strong and somewhat impractical assumption, especially in RLHF or drug discovery, where the true reward is unknown and expensive.

3. The offline assumption avoids the exploration problem, but the datasets still seem to come from a "behavior policy." The paper does not clarify how sensitive reward selection is to dataset coverage or distribution shift.

**Questions:**

1. Can the authors formalize conditions (e.g., approximate submodularity) under which the sequential-greedy method is guaranteed to achieve a bounded approximation to the optimal reward selection?  The notion of "which rewards matter" remains qualitative. What precisely defines a useful state, e.g., maximal marginal improvement in return, uncertainty reduction, or information gain? Without a formal criterion, the selection strategies risk degenerating into heuristic subset search. A theoretical grounding (e.g., influence-function or information-theoretic analysis) would make the contribution more solid.

2. How does this differ from Active Preference-Based Reward Learning?

3. The paper treats unlabeled samples as having zero (or minimal) rewards, following the UDS baseline. However, if high-value states are systematically unlabeled, this assumption introduces a structural bias in the Bellman backups and may lead to overly conservative policies.

4. Could reward selection be cast as an information gain problem over state-action pairs?

5. How realistic is the assumption of an evaluator providing exact policy returns?

6. In RLHF, one cannot compute expected returns directly, only obtain relative preference judgments. How would the framework adapt if evaluator feedback is pairwise, noisy, or delayed?

7. For large-scale LLM alignment, the state space is on the order of 10e+10 - 10e+12. How would the proposed strategies (especially sequential-greedy or ES) scale?

8. Could approximate embeddings or surrogate models be used to guide reward selection?

9. Since the authors use the UDS algorithm that imputes missing rewards as zeros, does this create bias in Q-function estimation?

10. The results show strong domain dependence of heuristics. Could the authors quantify what structural properties (e.g., reward sparsity, connectivity, stochasticity) determine which heuristic is optimal?

11. The reward selection is formulated as a static one-shot decision over an offline dataset. In practice, however, labeling decisions are often adaptive, which means the set of informative states depends on the evolving policy. How would the proposed framework extend to an iterative or online selection process? Would previously "unhelpful" states become valuable after the policy improves?

---

> ### Author Response · Authors · 2025-11-20
>
> Thank you for your review! We appreciate your engagement with the paper and recognition of its contributions.  We address the weakness and questions below and hope they address your concerns.
>
> > Theoretical guarantees for reward selection
>
> We appreciate the interest in a deeper theoretical characterization of reward selection. Providing nontrivial approximation guarantees or optimality-gap bounds in our setting is, however, extremely challenging. Such an analysis would require strong, idealized assumptions that do not hold in the *general offline RL setting* we consider. A formal theoretical analysis would need to bound the performance of (1) selection strategy: which is a complex function of the transition function, (partial) reward function and the output of the policy optimization process (e.g., Q-learning or IQL), and (2) that of the optimal strategy. To our knowledge, no existing theoretical framework captures these challenges. A theoretical treatment may be feasible under a significantly simplified setup, for instance, analyses such as that in PEVI [1] demonstrate the level of structural assumptions required to obtain guarantees in offline RL.
> Our goal in this work is to *introduce and formalize the reward selection problem* and to provide *a systematic empirical analysis* of its structure. The patterns observed in optimal and near-optimal state subsets selected under limited feedback suggest promising directions for future theoretical work, but developing such theory is beyond the scope of this initial formulation. To ensure generality of the empirical analysis, we intentionally study domains that span a wide range of traits: sparse rewards, bottleneck states, deterministic and stochastic transitions, and large multidimensional state spaces.
>
> > "oracle evaluator" in practice
>
> Indeed, knowledge of the true reward function is rarely feasible in practice. For this reason, a standard proxy for approximating the true performance of a policy is the online deployment of the policy. For the example of AI-driven drug discovery, this corresponds to experimentally assessing the effectiveness of a candidate molecule and its associated costs such as: lab work, human hours and monetary costs. The estimate of the policy's performance thus obtained is what we model as the oracle evaluator in our setup.
>
> > Dependence on behavior policy
>
> This is a great point! We study precisely this dependence of the reward selection strategy on the behavior policy in Appendix D.6. It is observed that *sufficient dataset coverage* provided by the behavior policy is the dominant factor in the final policy performance. The specific performance of the behavior policy matters lesser, i.e., a high performing policy can be learnt from data collected from a low performing policy with sufficient coverage, indicating that *distribution shifts* do not significantly impact reward selection, data coverage primarily does.
> Thank you for highlighting this point, we have updated the manuscript to emphasize the reference to this analysis [Line ~398 in blue].
>
> # Questions
>
> Next, we answer the questions one-by-one.
>
> >  Can the authors formalize conditions (e.g., approximate submodularity) under which the sequential-greedy method is guaranteed to achieve a bounded approximation to the optimal reward selection? The notion of "which rewards matter" remains qualitative. What precisely defines a useful state, e.g., maximal marginal improvement in return, uncertainty reduction, or information gain? Without a formal criterion, the selection strategies risk degenerating into heuristic subset search. A theoretical grounding (e.g., influence-function or information-theoretic analysis) would make the contribution more solid.
>
> We address this under "Theoretical guarantees for reward selection" above.
> Regarding what precisely defines a useful state: the utility of a state cannot be defined by itself and strongly depends on the states that were selected before it. The utility of a *state subset* is defined by the overall performance improvement: captured by $P(\pi_{[S_{[B]}]}) - P(\pi_{[S_{[0]}]})$, where $P(\pi_{[S_{[0]}]})$ defaults to the performance of the uniform-random policy.
> In sequential-greedy, we approximate this with a greedy per-step objective  $\Delta (s|S_{[b−1]}) = P(\pi_{S_{[b−1] \cup {s}}}) − P(\pi_{S_{[b−1]}})$ that can be seen as defining the *conditional utility* of a state. This greedy maximization is indeed expected to be optimal when a certain notion of *submodularity* holds as you correctly mention: however, the specific conditions on the reward and transition function required for such submodularity to hold for RL, to our knowledge, have not been studied. We would appreciate references, if any, to that end.

---

> > ### Author Response · Authors · 2025-11-20
> >
> > >  How does this differ from Active Preference-Based Reward Learning?
> >
> > As you highlight in point #1 under **Strengths**, methods such as Active Preference-Based Reward Learning [2] and other reward learning methods focus on reward estimation rather than policy performance, typically using function approximation and assumptions such as linearity of reward functions. We reference this and other works and highlight this difference under Related Works.
> >
> > >  The paper treats unlabeled samples as having zero (or minimal) rewards, following the UDS baseline. However, if high-value states are systematically unlabeled, this assumption introduces a structural bias in the Bellman backups and may lead to overly conservative policies.
> >
> > Yes, UDS’s zero‑imputation is intentionally biased: unlabeled states are treated as having minimal reward which implicitly nudges policies to gravitate towards states with *known rewards*. This can be thought of as pessimistic or conservative updates that is standard in offline RL, to ensure learning from observed (known) data rather than extrapolated (unknown and potentially erroneous) data. Alternatively, the known reward-labeled data can be viewed as "expert data" towards which the policy gravitates when in regions with unknown rewards.
> >
> > >  Could reward selection be cast as an information gain problem over state-action pairs?
> >
> > That is possible. Although, this information gain would need to be defined in terms of the unknown reward function, to capture a term akin to $\Delta (s|S_{[b−1]})$ which may be infeasible. This would differ from standard information gain formulations that aim to mitigate *uncertainty*.
> >
> > >  How realistic is the assumption of an evaluator providing exact policy returns?
> >
> > Discussed under ""oracle evaluator" in practice".
> >
> > > In RLHF, one cannot compute expected returns directly, only obtain relative preference judgments. How would the framework adapt if evaluator feedback is pairwise, noisy, or delayed?
> >
> > The generality of the framework allows it to be agnostic to the specific format of the evaluative feedback. As long as pairwise or noisy judgements can be converted to some scalar value (typically for RLHF that happens via the BTL model) that the evaluator can provide, this framework applies as is.
> >
> > > For large-scale LLM alignment, the state space is on the order of 10e+10 - 10e+12. How would the proposed strategies (especially sequential-greedy or ES) scale?
> >
> > sequential-greedy would be very impractical for a state space that large, given its training cost of $O(B|S|)$. ES will scale more feasibly, since it's training cost $O(km)$ is determined by k and m only, and not by the size of the state space.
> >
> > > Could approximate embeddings or surrogate models be used to guide reward selection?
> >
> > We are unclear about what approximate embeddings or surrogate models refer to. Please could you elaborate?
> >
> > > Since the authors use the UDS algorithm that imputes missing rewards as zeros, does this create bias in Q-function estimation?
> >
> > Discussed under Question #2 above.
> >
> > > The results show strong domain dependence of heuristics. Could the authors quantify what structural properties (e.g., reward sparsity, connectivity, stochasticity) determine which heuristic is optimal?
> >
> > For domains with restricted connectivity (bottleneck states), uniform sampling tends to be effective since it has a higher likelihood of sampling relevant yet rarely visited states. For sparse reward setups, visitation based sampling works well, particularly at low budgets, and those benefits also naturally extended to guided sampling. Overall, it matters that pathways along with reward information flows in a specific domain get selected. We have added and highlighted these points in the updated version (Section 4.1 and 4.3).
> >
> > > The reward selection is formulated as [...] Would previously "unhelpful" states become valuable after the policy improves?
> >
> > We completely agree that labelling must often be iterative and adaptive. For that exact reason, several of the strategies we study, such as guided, the on-policy variants, and sequential-greedy, select states iteratively, updating their choices based on partial progress (see Appendix C: Table 2 where we highlight the iterative vs one-shot (batch) strategies). The reward selection framework thus incorporates this aspect.
> > As for whether previously “unhelpful” states could later become valuable: absolutely. And this is also the reason why a submodularity like condition for sequential-greedy may at times not hold.
> >
> > We appreciate your deep engagement with our work. We hope the responses above address your concerns and questions, and we would be glad to clarify any remaining points if helpful.
> >
> > ---
> > [1] Jin, Ying, Zhuoran Yang, and Zhaoran Wang. "Is pessimism provably efficient for offline rl?." International conference on machine learning. PMLR, 2021.
> >
> > [2]  Sadigh, Dorsa, et al. Active preference-based learning of reward functions. 2017.

---

> > > ### Comment · Reviewer_XCoh · 2025-11-27
> > >
> > > Thank you for your response. You have addressed most of my questions; however, my main concern remains unresolved. I understand that some work is primarily empirical, but in such cases it is common to provide extensive experiments to demonstrate effectiveness—four simple Gym environments feels insufficient. If there were strong theoretical support and the experiments were only meant to validate the theory, this experimental scale would be more acceptable. I originally asked about the scalability of the approach, and as you have acknowledged, the method does not appear to scale to larger models.
> > >
> > > Also, in your response you stated that “the utility of a state cannot be defined by itself and strongly depends on the states that were selected before it.” To me, this seems to suggest that the Markov property no longer holds, since in a standard MDP the value of a state should depend only on the current state (and action), not on the full past trajectory. Under your formulation, the problem is in fact no longer an MDP.  And in such a setting, how do you justify or guarantee the convergence of the proposed training procedure?

---

> > > > ### Author Response · Authors · 2025-11-28
> > > >
> > > > > provide extensive experiments
> > > >
> > > > We agree that empirical analysis must be conducted on (1) a diverse set of domains, (2) at large scales, and (3) with sufficient statistical significance. As we highlighted in our response, our experiments are designed to meet the above criteria, and certainly go beyond a small set of Gym domains: we evaluate on MinAtar domains that have state spaces to the order of $10^5$ (Table 10). All training-free selection methods readily scale to such domains.
> > > >
> > > >
> > > > >  the value of a state should depend only on the current state (and action), not on the full past trajectory
> > > >
> > > > The sequential decision-making domains we study the problem of reward selection on are standard MDPs: the environment dynamics and returns satisfy the Markov property. Our remark that “the utility of a state … depends on the states that were selected before it” refers specifically to the *selection objective*: the marginal value of adding a particular state to the set of already selected states naturally depends on which states are already in that set.
> > > >
> > > > Regarding convergence, selection methodologies just pick $B$ states out of a possible $|S|$. Therefore, we do not associate a notion of convergence with such a problem where every method always picks some $B$ states. Could you please elaborate on what notion of convergence you have in mind, and which training procedure you are referring to, so that we can address this concern more directly?

---

### Official Review · Reviewer_DKrd · 2025-10-31

**Soundness:** 2
**Presentation:** 3
**Contribution:** 2
**Rating:** 2
**Confidence:** 4

**Summary:**

This paper defines a problem in offline reinforcement learning: given a fixed dataset, how can one select a subset to label with rewards so that the resulting policy achieves the best performance? The authors explore several data selection strategies and evaluate their feasibility, performance, and computational cost.

**Strengths:**

1. The paper addresses a common practical issue: selecting a useful subset from a large dataset for training.
2. The problem formulation is clearly defined with mathematical expressions.
3. The paper is well organized and easy to follow.

**Weaknesses:**

1. The problem setting is straightforward, and the formalization is rather direct. Although the paper discusses brute-force, sequential-greedy, and evolutionary strategies, these approaches are standard and lack conceptual novelty. The work reads more like a comparative survey than a contribution of a new algorithmic idea.
2. The motivation centers on high labeling-cost scenarios such as LLMs or drug discovery, yet all experiments are conducted on standard RL environments where labeling is cheap. This weakens the paper’s claim that it addresses real high-cost labeling settings.

**Questions:**

1. How exactly does the sequential-greedy strategy select the (i)-th new state? Since each step requires maximizing $\Delta(s | S_{[b]})$, does this imply comparing all candidates in $D - S_{i-1}$?
2. For the three training-phase strategies, is the evaluation cost prohibitively high? Comparing two subsets $S_{[B]1}$  and $S_{[B]2}$ requires training a full offline RL policy, which could be extremely expensive in domains like LLMs. The cost of policy training might dominate the cost of data selection itself.
3. Why was the UDS algorithm (Yu et al., 2022) chosen as the offline RL backbone? It is not SOTA. Using a stronger recent ORL algorithm might change conclusions, especially under low feedback ratios. The observed performance gaps may stem from the baseline’s limitations rather than the selected data. In other words, maybe with different selected labeled data, a stronger ORL algorithm can achieve similar performance?
4. How does the proposed framework handle sparse-reward settings? When many states have originally zero rewards, selecting such states is ambiguous, i.e., are they effectively “selected” or not? Moreover, many modern ORL algorithms already handle sparse rewards well, so it is unclear whether reward selection still offers additional benefits in these cases.
5. The paper title and term “reward selection” (and “Which reward matters?”) may be misleading. The problem focuses on **data selection**, not on choosing among reward functions. A more precise term might improve clarity.

---

> ### Author Response · Authors · 2025-11-20
>
> Thank you for your review!
>
> We respond to weaknesses raised below and hope the clarifications address the reviewer's concerns.
>
> > Novelty of problem setting and formalization
>
> We respectfully disagree with the characterization of the problem setting as straightforward. To the best of our knowledge, the formulation of reward selection for offline RL under a limited reward-labeling budget has not been previously studied, and its importance and novelty have been acknowledged by **all other reviewers**.
> Consequently, a  key contribution of our work is the formalization of the reward selection problem itself  given both its practical importance and the fact that it constitutes an understudied area. The paper does a systematic and principled treatment of this problem, with empirical analysis that lays the conceptual framework and key insights to guide future research.
>
> > Experimental domains
>
> Our goal in selecting experimental domains is to ensure diversity of structural characteristics while remaining in environments where extensive experimentation is feasible. The domains we consider serve a didactic purpose, and we intentionally chose ones spanning a wide range of traits: sparse rewards, bottleneck states, deterministic and stochastic transitions, and large multidimensional state spaces. This diversity is essential for understanding how different selection strategies behave under distinct structural conditions.
> Although the broader motivation for studying reward selection comes from high-cost labeling domains, effectively comparing strategies requires tractable settings where we can run a *sufficient number of trials to ensure statistical significance* in our findings. Scaling the analysis and strategies to larger, real-world domains is a natural and exciting direction for future work, and we view our contribution as providing the necessary foundation for such developments.
>
> # Questions
> Next, we answer the questions one-by-one.
>
> > How exactly does the sequential-greedy strategy select the (i)-th new state? Since each step requires maximizing $\Delta(s | S_{[b]})$, does this imply comparing all candidates in $D - S_{i-1}$?
>
> Yes, at iteration $b$, sequential‑greedy evaluates every candidate $s \in S \setminus S_{[b−1]}$. Note that this is the set of states not selected thus far, and *not* all the remaining datapoints in $D$.
> Each candidate is added to $S_{[b−1]}$ as $S_{[b−1]} \cup {s}$, and the resulting policy is used for computing $\Delta (s|S_{[b−1]}) = P(\pi_{S_{[b−1] \cup {s}}}) − P(\pi_{S_{[b−1]}})$. The best $s$ is added to $S_{[b]}$. This yields $O(B|S|)$ calls to the evaluator (Section 3.2).
>
> > For the three training-phase strategies, is the evaluation cost prohibitively high? Comparing two subsets $S_{[B]1}$ and $S_{[B]1}$ requires training a full offline RL policy, which could be extremely expensive in domains like LLMs. The cost of policy training might dominate the cost of data selection itself.
>
> Great point! The evaluation cost is indeed prohibitively high, particularly for sequential-greedy and brute-force (as highlighted in Figure 3). In this work, these strategies served as analytical tools to benchmark the highest attainable performance and showcase the gap with training-free strategies.
>
> > Why was the UDS algorithm (Yu et al., 2022) chosen as the offline RL backbone? It is not SOTA. Using a stronger recent ORL algorithm might change conclusions, especially under low feedback ratios. The observed performance gaps may stem from the baseline’s limitations rather than the selected data. In other words, maybe with different selected labeled data, a stronger ORL algorithm can achieve similar performance?
>
> Our setting differs fundamentally from the traditional offline RL setting, where every transition is assumed to have a known reward. In our problem, only a subset of states may have reward labels, and the remainder of the dataset contains no reward information at all. Many recent high-performing offline RL algorithms are designed specifically for fully labeled datasets and are therefore not applicable in our partially labeled regime. UDS (Yu et al., 2022) is one of the few methods that explicitly studies learning policies from partially labeled data, making it appropriate for our RLLF setup. For this reason we adopt UDS as the base policy-learning algorithm in our main experiments.
> To verify that our conclusions are not an artifact of UDS, we also implement an alternative value-based RLLF algorithm -- Adaptive Q-learning -- in Appendix D.9, which handles unlabeled states by truncating Q-updates. As seen by the results in Tables 15–16, the same conclusions as described in the main text hold.

---

> > ### Author Response · Authors · 2025-11-20
> >
> > > How does the proposed framework handle sparse-reward settings? When many states have originally zero rewards, selecting such states is ambiguous, i.e., are they effectively “selected” or not? Moreover, many modern ORL algorithms already handle sparse rewards well, so it is unclear whether reward selection still offers additional benefits in these cases.
> >
> > Our empirical analysis covers several sparse-reward domains--TwoRooms, FrozenLake, TwoRooms-Trap, and CliffWalk--in which the only positive reward is at the goals and the latter three also include trap or cliff states with negative rewards. These environments reflect the sparse-reward setting the reviewer highlights.
> > Reward selection for sparse reward domains is still beneficial when the domain has a large state set--consider the example of LLMs. In such cases, if it is *known a priori* that the reward function is sparse, then the search space of reward selection can be drastically reduced by constraining selection to only states with non-zero rewards: thereby increasing the effectivity of the selection method.
> >
> > > The paper title and term “reward selection” (and “Which reward matters?”) may be misleading. The problem focuses on data selection, not on choosing among reward functions. A more precise term might improve clarity.
> >
> > The question may be stemming from a misinterpretation related to the first question: regarding the difference between selecting a datapoint versus selecting the reward at a specific state.
> > All of the data is available beforehand, which of the data to label (as identified by the states that constitute the desired datapoints) and with what (specific reward values at those states) is the central question, and hence the choice of the phrase "reward selection". The policy learned from a partially labeled dataset is shaped by the states at which rewards are known, and more importantly, the specific values of rewards at those states.
> > This is a subtle point, we thank the reviewer for highlighting that we can make it clearer in the updated version.
> >
> > We hope these clarifications address your concerns. Please let us know if any further details would be helpful.

---

### Official Review · Reviewer_QqrC · 2025-11-01

**Soundness:** 3
**Presentation:** 2
**Contribution:** 3
**Rating:** 6
**Confidence:** 2

**Summary:**

This paper introduces and formalizes the problem of Reward Selection for Reinforcement Learning from Limited Feedback (RLLF). The paper asks: given a fixed budget B for reward labels, which subset of states from an offline dataset should be labeled to maximize the performance of the final policy?

The authors use an offline RL setup. A reward selection strategy $\mathcal{Q}^{(B)}$ selects a state subset to be labeled. A policy is then learned from the resulting partially-labeled dataset. The paper investigates two categories of strategies (1) heuristic strategies (no training): and (2) strategies with a training phase

Experiments show that heuristic performance is domain-dependent. The authors conclude by identifying structural patterns of optimal state sets, such as prioritizing "anchor" states on optimal paths and critical penalty states.

**Strengths:**

- The paper formalizes the offline RLLF problem
- The problem is well-motivated by high-impact, real-world applications where feedback is the primary bottleneck
- The paper has a very thorough experimental validation

**Weaknesses:**

- It is not entirely clear how the on-policy methods compute the state-visitation distributions. Could the author clarify if they assume the full transition dynamics to be known in order to compute the visitations?
- The "training phase" seems to be quite computationally expensive, given that for every additional labelled feedback, it requires retraining a policy. Could the authors elaborate on this point and the justification for this cost?
- While it is true that the cost analysis for EM does not depend on S and B, I found it slightly confusing to present only k and m as parameters chosen by the user. Optimal k and m are probably still functions of S and B and the evolutionary strategy used to update $\theta$. Is there any way to relate k and m with S and B? This would give a more direct comparison.

**Questions:**

See Weaknesses

Additionally, could the authors please clarify how the state-visitation distribution $d^{\pi_{[b-1]}}$ for the updated policy is estimated from a fixed offline dataset? Does this approach assume access to a model of the dynamics?

---

> ### Author Response · Authors · 2025-11-20
>
> Thank you for your review! We appreciate your positive assessment of our work, and particularly appreciate the summary which is accurate in every detail.
>
> We respond to questions raised under weaknesses below and hope they address your concerns.
>
> > State visitation distribution computation in on-policy selection methods
>
> In our experiments, the ground-truth transition dynamics are *not assumed to be known*, and we do not assume access to the a model of the environment. For the on-policy methods, we construct a maximum likelihood estimate of the transition model directly from the offline dataset by empirically counting the number of transitions, i.e., $\widehat{p}(s, a, s') = N(s, a, s’) / N(s, a)$. This estimated model is used to compute the state visitation distribution for the policy under consideration, in this case $\pi_{[b-1]}$. The estimated transition model $\widehat{p}$ naturally has finite sample estimation error which consequently propagates into the computed visitation distribution. Nevertheless, our empirical results show that even with this approximate model, the on-policy methods are effective in several domains (Table 1).
>
> We appreciate the opportunity to clarify these implementation details of the on-policy methods, and have made them more explicit in our updated version [Lines 252-255].
>
> > Training phase is computationally intensive due to policy retraining
>
> Thank you for raising this point, it raises an important subtlety in how we define the **cost** in the training phase. The central motivation of the work is that obtaining reward feedback is a practical bottleneck. Accordingly, the cost captures the operational cost of **evaluating a policy** (in terms of the number of calls to the evaluator $\Xi$) rather than the computational cost of retraining a policy (under RLLF). A helpful analogy is to view an evaluator call as the online deployment of a policy to measure its performance. For the example of AI-driven drug discovery, this corresponds to experimentally assessing the effectiveness of a candidate molecule and its associated costs such as: lab work, human hours and monetary costs.
> The feedback about the performance of the policy thus obtained can be used to refine the selection methodology and retrain the policy. The retraining certainly bears computational costs, but in settings where the main concern is minimization of the costs of obtaining evaluative feedback (rewards), the relatively manageable computation cost of retraining is justified.
> We have added a comment regarding this in the updated version [Line 264].
>
> > Optimal choice of k, m for ES as functions of S and B
>
> ES as an optimization strategy permits mainly two parameters, $k$ and $m$, which determine the number of iterations (k) and the population size per iteration (m). The quality of the optimization of the selection strategy depends entirely on these two parameters. While one could imagine heuristic rules-of-thumb for choosing them—for example, selecting $k$ and $m$ such that $k×m \approx f(S,B)$, as the reviewer suggests—our empirical analysis indicates that the relationship is more complex and likely depends on properties of the transition and reward structure of the domain: as the reviewer correctly highlights in the summary.
> Our ablation studies in Tables 6 and 7 reinforce this point: the effective choice of $k$ and $m$ is a non-trivial function of several factors, including the size of the state set, the complexity of the reward and transition structures (e.g., domains with bottleneck states versus more uniform ones), and the label budget $B$. These results suggest that any rule governing $k$ and $m$ would necessarily depend on problem-specific structure.
>
> We hope these clarifications address your concerns. Please let us know if any further details would be helpful.

---

### Author Response · Authors · 2025-11-26

With a week remaining in the discussion period, we hope that our responses have helped address the original concerns. If any reviewers have additional comments or updated perspectives after reading our responses, we would be eager to have a discussion.

---

### Meta-Review · Area_Chair_buMK · 2026-01-06

**Summary:**

This paper studies reward selection for reinforcement learning under limited feedback, formalizing the problem of choosing which states in an offline dataset should be labeled with rewards to maximize downstream policy performance. While the problem is clearly motivated and the empirical study is thorough, I do not believe the submission meets the bar for publication due to limited novelty and insufficient differentiation from existing literature. Conceptually, the proposed setting is closely related to well-established work on active learning, machine teaching, and active reward learning, where the central question is how to select informative feedback under a budget. In particular, extensive prior work has already addressed active reward acquisition in online RL (a strictly harder setting due to exploration constraints) as well as active reward function learning in offline settings, which is nearly identical to the formulation here except that the objective is reward estimation rather than direct policy optimization—a distinction that does not fundamentally change the structure of the problem, since reward estimation and policy optimization are tightly coupled. Despite this close relationship, the paper does not meaningfully engage with or build upon this literature, nor does it leverage its theoretical tools (e.g., submodularity, information gain, teaching dimension). Methodologically, the paper introduces no new algorithmic ideas beyond straightforward baselines such as greedy selection, evolutionary search, and simple heuristics, and the training-based strategies rely on strong oracle assumptions without offering scalable or principled alternatives. As a result, the contribution is largely empirical and comparative, without delivering new conceptual or algorithmic insights that would justify publication at this venue. Overall, while the work is carefully executed and the problem is important, I do not find that it advances the state of the art sufficiently beyond existing active learning and reward learning frameworks to warrant acceptance.

**Reviewer Scores:**

NA

---

### Decision · Program_Chairs · 2026-01-26

Reject